# Where Do Large Learning Rates Lead Us?

**Ildus Sadrtdinov**[1,2]*, **Maxim Kodryan**[2]*, **Eduard Pokonechny**[3]*,
**Ekaterina Lobacheva**[3]†, **Dmitry Vetrov**[1]†
[1] Constructor University, Bremen    [2] HSE University    [3] Independent researcher
isadrtdinov@constructor.university, mkodryan@hse.ru, epokonechnyy@gmail.com
lobacheva.tjulja@gmail.com, dvetrov@constructor.university

## Abstract

It is generally accepted that starting neural networks training with large learning rates (LRs) improves generalization. Following a line of research devoted to understanding this effect, we conduct an empirical study in a controlled setting focusing on two questions: 1) how large an initial LR is required for obtaining optimal quality, and 2) what are the key differences between models trained with different LRs? We discover that only a narrow range of initial LRs slightly above the convergence threshold lead to optimal results after fine-tuning with a small LR or weight averaging. By studying the local geometry of reached minima, we observe that using LRs from this optimal range allows for the optimization to locate a basin that only contains high-quality minima. Additionally, we show that these initial LRs result in a sparse set of learned features, with a clear focus on those most relevant for the task. In contrast, starting training with too small LRs leads to unstable minima and attempts to learn all features simultaneously, resulting in poor generalization. Conversely, using initial LRs that are too large fails to detect a basin with good solutions and extract meaningful patterns from the data.

## 1 Introduction

Understanding neural networks (NNs) training is one of the major goals in deep learning for various reasons, from developing more efficient training protocols to handling the AI safety issues [10]. Unfortunately, the high-dimensionality, non-convexity, and stochasticity of the optimization process make its comprehensive analysis extremely difficult. Learning rate (LR) is perhaps the most important hyperparameter in a gradient descent optimizer [11]. LR controls the optimization step size and, due to extreme non-convexity of the optimized loss function yielding a manifold of qualitatively different minima in the loss landscape of neural networks, this hyperparameter is primarily responsible for the type of solution we obtain after training [5, 6, 35, 41, 60, 62, 68].

Using large learning rates, especially at the beginning of training, has become a common practice [23, 38, 69]. Starting with large LR values is known to help avoid poor local minima [13, 34, 40, 48] while the solutions obtained at the end of training often have favorable properties like good generalization and flatter loss landscape around the corresponding optima [25, 35, 56, 60]. Prior work has attempted to explain the benefits of high learning rates from various perspectives, which can be divided into three main areas: optimization [5, 9, 27–29, 58, 59, 62, 65], model sparsity [2, 6, 12, 51, 55], and pattern learning [41, 44, 57, 68]. Nevertheless, the following questions still remain very relevant:

1. *Large LRs are preferred but how large are we talking about?*
2. *What are the key characteristics of the models trained with different LRs?*

---

*Equal contribution.
†Shared senior authorship.

Concerning the first question, a general recommendation is to start training with an LR value that is too high for convergence but too low for divergence [5, 6, 25, 35], with no precise benchmarks available over this wide range. Concerning the second question, despite the abundance of literature on the role of the LR hyperparameter in NN training, we still lack a unified picture of how exactly models trained with small and large LRs differ.

We conduct a detailed empirical analysis on both of the above-mentioned problems in a special setting that allows for more precise control of the LR value. We structure our analysis around the taxonomy of Kodryan et al. [35] who classified different LR values into three training regimes: 1) convergence for small LRs, 2) chaotic equilibrium for medium LRs, and 3) divergence for large LRs. We start with examining the generalization of the final solutions obtained by either fine-tuning with a small LR or weight averaging after training in different regimes. We confirm that the best choice is to start training in the second regime, with moderately high LRs that lead to neither convergence nor divergence, but only a relatively narrow portion of that range, which we define as *subregime 2A*, consistently provides optimal results. To investigate the effectiveness of these learning rates, we examine training in different regimes from both loss landscape and feature learning perspectives. First, we explore the local geometry of the reached minima and reveal that training in subregime 2A locates a basin in the loss landscape containing linearly connected well-generalizing minima. In contrast, using too small LRs can result in models that become unstable when fine-tuned with larger LRs, while using too large LRs fails to detect basins of good solutions. Then, to study feature learning in different regimes, we introduce a synthetic example with interpretable features. We find that as LR increases up to subregime 2A (including it), models become more specialized in the sense that they rely on fewer features in the data, while further increase in LR gradually impairs the ability to extract features from the data. We also show how our findings transfer to a practical image classification setting via frequency analysis of the input images. In sum, **our contributions** are as follows:

1. We compare the generalization of the minima reached after training with different initial LR values and find that the best choice is indeed to take LRs above the convergence threshold. However, only a relatively narrow part of this range, which we call subregime 2A, consistently provides optimal final results.

2. We discover that starting training with LRs from subregime 2A locates a convex basin containing optimal solutions that can be achieved via fine-tuning with a small LR or weight averaging. Too small initial LRs, in contrast, find unstable local minima, while too large LRs fail to locate such basins of good solutions.

3. We reveal that NNs tend to learn sparser set of the most relevant features from the data as the initial LR increases until the end of subregime 2A; after that, the model gradually loses ability to learn useful features.

4. We show that conclusions obtained in a special setting with accurate control of the LR hold for conventional neural network training as well.

Our code is available at `https://github.com/isadrtdinov/understanding-large-lrs`.

## 1.1 Related Work

A significant amount of theoretical and empirical deep learning research has been dedicated to studying the training dynamics of neural networks. Most of it concerns the role of the LR hyperparameter in NN optimization and generalization accentuating the favorable effects of large LR training.

A vast line of works attributed these effects to amplifying the magnitude of Stochastic Gradient Descent (SGD) noise, which effectively does not allow the model to converge to suboptimal local minima with high local curvature, or *sharpness* [27–29, 34, 58, 65]. Similar directions highlight other attributes of (stochastic) gradient descent training, such as implicit regularization [9, 19, 30, 59] or minima stability [46, 50, 52, 51, 62, 61, 63], enhanced by the use of large learning rates. While suggesting possible mechanisms for why large LR values lead to good solutions, these works still lack characterization of these solutions and practical receipts for finding them.

A closer look at training with large learning rates reveals its tendency to favor simpler and *sparser* solutions. Specifically, Andriushchenko et al. [6] demonstrated that hovering at some constant loss level when training with large LRs helps optimization to eventually find modes with sparse activation patterns in hidden layers of deep neural networks. Similarly, Chen et al. [12] theoretically predicted

and empirically confirmed an increase in sparsity of neurons of networks trained with larger LRs. A recent work of Ahn et al. [2] discovered that large LR values are necessary to learn the "threshold neurons" in networks with ReLU activation, which effectively lead to sparser solutions. Sparsity bias with increasing LR has also been theoretically studied for linear diagonal [51] and two-layer ReLU [55] networks. We explore training with different initial LRs from the perspective of input space *feature sparsity*. In contrast to previous results that reported a monotonic increase in sparsity with increasing LR, we found that feature sparsity behaves non-monotonically: the most pronounced feature sparsity effect is observed at approximately the same initial LR values that provide the best final solutions in terms of generalization.

Another relevant research direction examines pattern learning with different learning rates. Typically, in specific artificial settings, these works show that more complex patterns are learned with smaller LRs, thus, to avoid overfitting, it is beneficial to start training with a large LR and then decay it to smaller values after learning all the "easy-to-fit" patterns from the data [41, 44, 68]. Recently, Rosenfeld and Risteski [57] suggested an intriguing idea of "opposing signals" referring to similar patterns in objects of different classes such as blue background in images of planes and ships. Opposing signals usually correspond to spurious features, which lead to suboptimal quality if learned by the model. As suggested by the authors, training with large LR values resolves opposing signals via filtering out the corresponding unreliable features. We follow this direction and provide further clarity on what features in the data the model captures after training with different initial LRs.

## 2 Methodology

To rigorously study the impact of the initial LR on the final solution, we require to fix it at the beginning of training. However, as most modern architectures use normalization in some form, truly fixing an LR becomes a nontrivial action. Specifically, scale-invariance induced by normalization yields two consequences: 1) scale-invariant weights are essentially defined on the sphere, and 2) the *effective learning rate* (ELR) of these weights, i.e., learning rate on the unit sphere, is varying even with a fixed LR due to a varying parameters norm [5, 7, 35, 42, 43, 53]. Therefore, to resolve this issue, we train our models with a fixed parameters norm in a scale-invariant manner, which helps us conduct our study more accurately since fixing an LR now leads to a fixed ELR as well. Following the prior research [5, 35, 42], we make all our models fully scale-invariant (SI) by fixing the last layer and removing trainable affine parameters of normalization layers, and train them using projected SGD on a sphere of fixed radius. We return to a more conventional setting in Section 6.

Investigating the training of scale-invariant models on the sphere, Kodryan et al. [35] discovered that it typically takes place in one of three regimes depending on the LR value: 1) *convergence*, when parameters monotonically converge to a minimum, 2) *chaotic equilibrium*, when loss noisily stabilizes at some level, and 3) *divergence*, when a model has random guess accuracy. We adopt the three-regimes taxonomy and build our analysis around it from here on. In Figure 1, we show the (smoothed) test accuracy after training with different fixed LRs. The three regimes are clearly distinguishable. Kodryan et al. [35] mostly focus on training with fixed LRs, however, they point out that starting training in the second regime and then decreasing LR can often lead to better solutions than training with a constantly fixed LR. Similarly, other works [5, 6] suggest that training should start in the second regime and attribute this effect to the benign noise driven process happening in the "loss stabilization" phase.

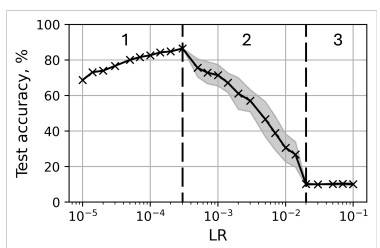

Figure 1: Three regimes of training with a fixed LR. Mean test accuracy $\pm$ standard deviation on the last 20 out of 200 epochs are shown. Dashed lines denote boundaries between the training regimes. SI ResNet-18 on CIFAR-10.

Following these results, we wish to analyze the points obtained after initial training with different LR values from the perspective of their utility for subsequent training with small LRs or weight averaging. To this end, we divide training into two stages. First, we perform so-called *pre-training*, i.e., we train models with different fixed LRs, which we call **P**LRs, for sufficient amount of epochs to ensure stabilization of training dynamics. After pre-training, we either 1) change the learning rate and *fine-tune* the model, i.e., train it further with a small LR, or 2) continue training with the same LR as at the pre-training stage and weight-average consequent checkpoints, as is usually done in stochastic

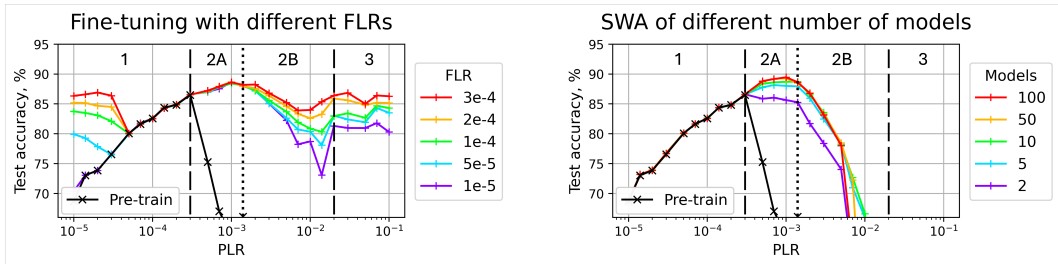

Figure 2: Test accuracy of the fine-tuned (left) and SWA (right) solutions for SI ResNet-18 on CIFAR-10. Test accuracy after pre-training is depicted with the black line. Dashed lines denote boundaries between the pre-training regimes, dotted line divides the second regime into two subregimes.

weight averaging (SWA) [26]. For fine-tuning, we use only first regime LRs, which we call **FLRs**, to ensure model convergence. Further detail on the experimental setup are provided in Appendix A.

In the following sections, we present our main results. All results are obtained with a scale-invariant ResNet-18 [23] trained on CIFAR-10 [37]. We additionally consider a plain convolutional network ConvNet and ViT small architecture [16] as well as CIFAR-100 and Tiny ImageNet [39] datasets in the appendix. For ease of presentation, we divide all plots into three parts (using dashed lines) corresponding to the three previously introduced pre-training regimes. We begin with comparing the generalization of the fine-tuned/SWA solutions obtained after pre-training with different LRs (Section 3). We then analyze their local geometry to shed more light on their differences (Section 4). Next, we shift our attention to studying feature learning in models trained with different initial LRs and propose a synthetic example with adjustable features (Section 5). Finally, we demonstrate that our findings remain valid for conventionally trained networks as well (Section 6) and draw conclusions.

## 3   Finding the best LRs for generalization

To examine the generalization benefits of the pre-training stage, we evaluate the test accuracy of the fine-tuned/SWA models. Figure 2 depicts the test accuracy after pre-training with different PLR values (black line) and after fine-tuning/SWA (colored lines). Below we successively analyze pre-training in each regime from the generalization point of view and highlight the range of initial LRs providing the best final quality.

**Regime 1**   In accordance with the results of Kodryan et al. [35], we observe a monotonic dependence between the PLR value and the test accuracy of the pre-trained model. Both SWA and fine-tuning with FLR ≤ PLR do not noticeably change the pre-trained accuracy. At the same time, fine-tuning with FLR > PLR can significantly improve the test accuracy, however, it still cannot provide a considerably better solution than training with the same FLR from scratch. Therefore, the best strategy of training in the first regime is to use the maximum constant LR without any schedules.

**Regime 2**   The second regime is the most promising for pre-training, since most practical LR schedules start in this regime [5, 6, 35]. We can see that the results strongly depend on the PLR value and the general advice "pre-train with a large LR to obtain a better solution" is valid, but definitely not for all PLRs of the second regime. In fact, the second regime can be divided into two subregimes, which we denote as **2A** and **2B**. We discovered that pre-training in subregime 2A, i.e., with lower second regime PLRs, results in significantly better fine-tuning and SWA results compared to regime 1, while pre-training in subregime 2B, i.e., with higher second regime PLRs, loses this advantage.

*Subregime 2A*   Even though pre-training with lower second regime PLRs does not converge to the lowest loss values, it locates optimal regions for further fine-tuning or weight averaging. Notably, fine-tuning a network obtained with a PLR from this range with *any* FLR results in minima of the same quality. So, such pre-training allows for fine-tuning even with small FLRs to avoid local optima with poor generalization, to which training from scratch with these FLRs usually converges. Moreover, the fine-tuned models are of higher quality than any solutions obtained in the first regime,

which shows that the best minima may be completely unreachable with small LRs from a standard initialization, at least in any reasonable training time.[3]

*Subregime 2B* However, further increases in PLR reduce the benefits of pre-training in subregime 2A. Both SWA and fine-tuning quality degrades; in addition, the solutions obtained with different FLRs begin to differ from each other in the test accuracy. So, such pre-training can be detrimental if we decrease LR to the first regime right after it. Previously, Andriushchenko et al. [5] also discovered that high LRs of the second regime can deteriorate final model performance for single-step LR schedules. However, we find that this effect can be mitigated by gradually decreasing an LR through subregime 2A (see Appendix D). In sum, obtaining high-quality results in this subregime using SWA or fine-tuning with a constant FLR is not possible and requires a more complex LR schedule.

**Regime 3** Pre-training in the third regime is somewhat similar to random walking in the parameters space [35, 53], hence it is completely impractical for weight averaging. Despite this, we notice that it still can be beneficial for fine-tuning with small FLRs. We conjecture that it is due to the fact that pre-training in the third regime, in contrast to standard random initialization [20, 22], yields a very uneven distribution of norms of individual scale-invariant parameter groups. Many groups have low norms implying that their effective learning rate is higher than the total ELR for the whole model, which promotes convergence to better optima during fine-tuning (see Appendix E).

**Takeaway 1** *Although pre-training with large first regime PLRs finds points with relatively high test accuracy, they cannot be improved via fine-tuning or weight averaging; hence, the first regime is not the best choice for starting training. Pre-training with PLRs of the lower part of the second regime, just above the boundary between regimes 1 and 2, helps robustly increase the quality to a level unreachable with a constant LR; however, using higher PLRs loses this advantage and degrades performance. Despite pre-training in regime 3 does not seem to extract any useful information from the data, it may help in subsequent fine-tuning by providing better initialization for the optimization.*

## 4 Loss landscape perspective

In the previous section, we determined the range of initial LRs leading to the best model quality. We now aim to clarify the key characteristics that differentiate these LRs from others. In this section, we take the loss landscape perspective and analyze the local geometry of minima obtained after initial training in different regimes. To do this, we measure the linear connectivity and the angular distance between them. Angular distance between two networks with weights $\theta_1$ and $\theta_2$ is calculated as

$$\angle(\theta_1, \theta_2) = \arccos\left(\frac{\langle\theta_1, \theta_2\rangle}{\|\theta_1\| \|\theta_2\|}\right).$$

We choose it as a natural metric on the sphere in the weight space. Linear connectivity is measured via calculating a linear-path barrier between two networks w.r.t. training or test error, i.e., the highest difference between the error on the linear path between two points in the weight space and linear interpolation of the error at each of them [17]:

$$B(\theta_1, \theta_2) = \sup_{\alpha \in [0,1]} \left[E(\alpha\theta_1 + (1 - \alpha)\theta_2) - \alpha E(\theta_1) - (1 - \alpha)E(\theta_2)\right],$$

where $\theta_1$ and $\theta_2$ are weights of the networks, and $E$ is the error measure. The barrier value shows whether solutions obtained from the same pre-trained point remain in the same low-error region (low barrier) or head to different optima (high barrier). In Figure 3, we present angular distances and linear connectivity between three solutions for each PLR: SWA of 5 networks and the points obtained after fine-tuning with the lowest and the highest considered FLRs.

As was shown previously, for small initial LR values attributed to regime 1 neither SWA, nor fine-tuning with smaller FLRs improve the pre-trained model. This is because they effectively remain in the same minimum: the obtained solutions are of similar quality, close to each other in angular distance and linearly connected. On the other hand, taking FLR $\gg$ PLR can cause a catapult effect [40] and subsequent convergence to a better minimum corresponding to the new LR value. This is clearly observed when fine-tuning with the highest FLR from low PLRs through the improvement

---

[3]We discuss this idea in more detail in Appendix C.

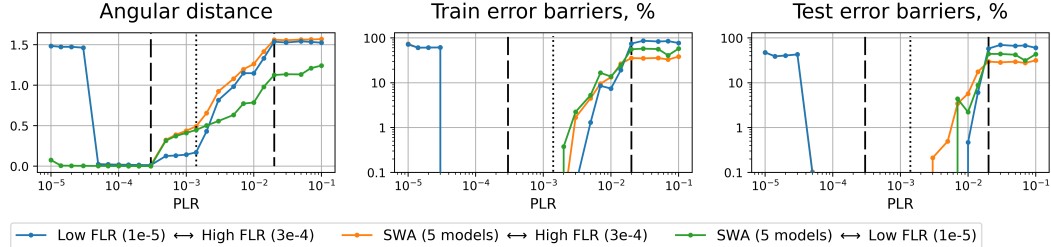

Figure 3: Geometry between the points fine-tuned with the smallest and the largest FLRs and SWA. SI ResNet-18 on CIFAR-10.

of test accuracy in Figure 2 and high angular distance and error barrier with other solutions in Figure 3. Thus, pre-training with small LRs leads to minima that have suboptimal generalization and are unstable in the sense that increasing LR after convergence can knock the model out of the current minimum and move it to a different one.

Moving to the right along the PLR axis, we encounter subregime 2A that is optimal for further weight averaging or fine-tuning: the resulting solutions are of high quality regardless of the FLR value. Geometrical inspection also reveals that the solutions obtained with different FLRs from the same pre-trained point are very close to each other in angular distance and linearly connected. SWA models are located farther from the fine-tuned solutions but still are mostly linearly connected to them.[4] Hence, we conjecture that pre-training in subregime 2A locates a "bowl" in the loss landscape by bouncing between its walls [64]; the bottom of this "bowl" is a convex basin of high-quality solutions, which can be easily reached via fine-tuning or weight averaging.

Pre-training with higher second regime PLRs gets stuck at higher loss levels and is unable to locate the region of high-quality solutions. With larger PLRs, fine-tuned solutions not only become worse but also differ from each other: we see a rapidly growing gap in test accuracy and angular distance between the solutions obtained with low and high FLRs, followed by a separation in the linear connectivity w.r.t. the training error. Interestingly, linear connectivity w.r.t. the test error is still preserved for the most part, which may be due to a high test error value at one end. We conclude that unlike subregime 2A, pre-training in subregime 2B explores a vast area of the loss landscape with a diverse set of minima.

Even though pre-training in the third regime can help fine-tuning with small FLRs converge to better minima thanks to the optimization effects, in many ways it still behaves as random walking with a large step size. Both the angular distances and the error barriers approach their upper limits (angular distance of $\pi/2$, which is a typical angular distance between two independent points in high-dimensional space, and random-guess error), completing the trend established in subregime 2B.

**Takeaway 2** *Pre-training with small PLRs attributed to the first regime can end up in unstable minima, from which the optimization escapes at increased FLRs. Using initial LRs from the optimal range (subregime 2A) allows for finding a basin that contains only high-quality solutions easily reachable via fine-tuning or SWA. Larger PLRs lose the ability to locate low-error basins and instead cover large areas of the loss landscape with diverse not linearly connected minima.*

## 5 Feature learning perspective

**Synthetic example**    We proceed to study the feature learning properties of models trained with different LRs. With this in mind, we propose a synthetic example with precise control over how different features affect the target variable. We consider a binary classification setting with the following three properties: 1) all three training regimes are observed; 2) varying the initial LR leads to different generalization; 3) the data points contain multiple features, each of which is sufficient to classify the data correctly. We use 32-dimensional data vectors, where each pair of coordinates

---

[4]Linear connectivity with SWA can be lost for more complex datasets like CIFAR-100, but this is expected since the SWA point is obtained by averaging several subsequent pre-training checkpoints, as opposed to fine-tuning from the same pre-trained model.

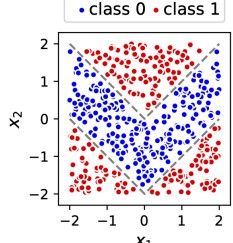

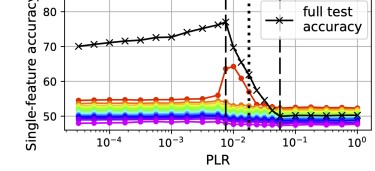

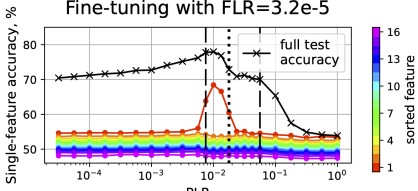

Figure 4: A single 2D "tick" feature used in the synthetic example.

Figure 5: Feature sparsification in the synthetic example for pre-training (left), and fine-tuning with FLR = $10^{-4}$ (right). Colored lines show the accuracy values on single-feature test samples, sorted independently for each training run. The accuracy on a regular test sample is depicted with the black line. The lines are averaged over 50 seeds.

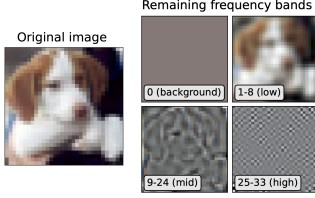

Figure 6: Inverse 2D DFT images, each containing 1 of 4 components of the spectrum. For this figure, we rescale each color channel of *low*, *mid* and *high* images to 0-1 range.

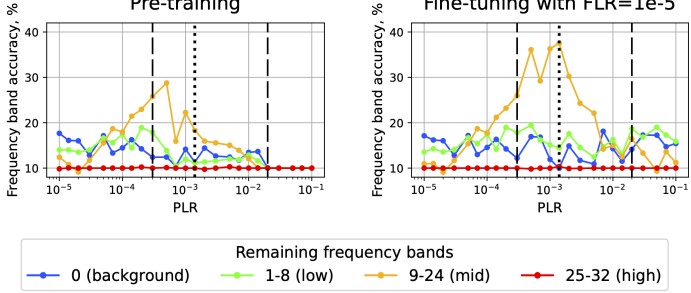

Figure 7: Accuracy of different frequency bands for pre-training (left) and fine-tuning with FLR = $10^{-5}$ (right). Each line is an average over 5 last epochs of training. SI ResNet-18 on CIFAR-10.

represents a single 2D "tick" feature (Figure 4), all 16 features are sampled from the same distribution. We use a 3-layer MLP with ReLU activation and Layer Normalization [8] and make it scale-invariant similarly to the main setup. We put further detail in Appendix A. The generalization and geometry properties in this setup are similar to those previously described, see Appendix G.

To quantify how features are learned with different LRs, we generate 16 single-feature test samples, each having only one of the 16 total features. The values of other features are distributed along the decision boundary (gray dashed line in Figure 4) to represent missing features. We measure the accuracy of the trained models on these single-feature test samples and relate the obtained values to the importance of the respective features for the model: the higher the accuracy value, the more important the feature. We analyze these values in sorted order for each PLR because models may favor different features in different training runs due to the randomness in initialization [3].

The results are depicted in Figure 5. For small PLRs, there is no feature selection: we observe similar accuracy w.r.t. different features resulting in poor overall generalization, since no target-predicting feature is reliably learned by the model. Closer to the boundary between regimes 1 and 2 we begin to observe a kind of model specialization: the accuracy corresponding to a single feature is significantly higher than for the rest. Moreover, such feature sparsity persists after fine-tuning, indicating completely different feature learning behavior than in regime 1: even in the setting of equally useful features, the model prefers to focus more on some subset of features instead of trying to learn all features at once. The sparsity peak in subregime 2A coincides with the peak of the fine-tuning test accuracy (Figure 15), confirming that a sparser set of learned features improves model generalization. This may also be related to the fact that the basin is determined exactly at this range of LRs (see discussion in Section 7). When the PLR is increased to subregime 2B, the ability to extract any useful patterns from the data is reduced, which manifests itself in degraded accuracy on both regular and single-feature test samples.

**Fourier features** A similar feature selection effect can be observed in scale-invariant ResNet-18 on the CIFAR-10 dataset. Since for the real-world image data it is generally not clear how to define features [57], we use frequency bands of the 2D Discrete Fourier Transform (DFT) as proxy for

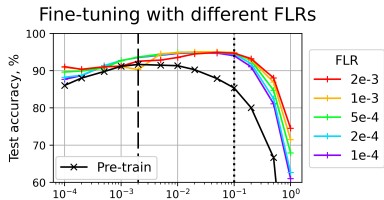 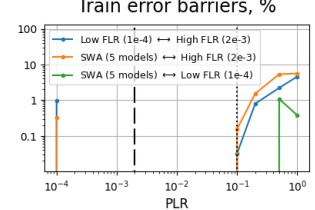 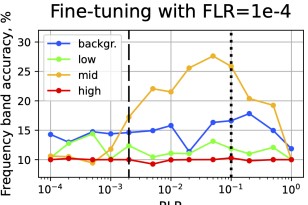

Figure 8: Practical ResNet-18 on CIFAR-10. Dashed lines denote boundaries between the first and second pre-training regimes, dotted line divides the second regime into two subregimes. Left: Test accuracy after fine-tuning with different FLRs (color lines) and after pre-training (black line). Middle: train error barriers between small FLR, high FLR, and SWA points. Right: Accuracy w.r.t. different frequency bands for fine-tuning with FLR $= 10^{-4}$.

features [1]. We divide the full 2D spectrum of an image into 4 components, each consisting of a range of frequency bands: 0 (constant background color), 1-8 (low), 9-24 (mid), and 25-32 (high). For each of the 4 components, we zero out the rest of the spectrum and apply the inverse DFT to obtain images with only one frequency component preserved (Figure 6) by analogy with the single-feature test samples in the synthetic example. We repeat this procedure with every test image and measure the accuracy on the resulting 4 new test sets corresponding to the spectrum components. A more detailed description of the setup can be found in Appendix A. See Appendix H for additional results.

Figure 7 shows the test accuracy w.r.t. each spectrum component after pre-training with different PLRs. As in the synthetic example, small PLRs of regime 1 tend to treat all features approximately equally, paying slightly more attention to the background color and low-frequency features, while increasing PLR introduces feature sparsity, making the mid-frequency features significantly more important. The peak of the mid-frequency accuracy is achieved in subregime 2A, reaching much higher absolute values than the 0 and 1-8 components in the first regime. Moreover, after fine-tuning, the mid-frequency accuracy improves even further, showing the same bias towards a subset of features as in the synthetic example. However, a further increase in the PLR to subregime 2B reduces the feature sparsity and the mid-frequencies importance. The mid-frequencies are known to play a key role in model generalization and robustness in image classification [1, 66], so their apparent prevalence in subregime 2A indicates that feature selection is not random but is biased towards the most useful features for the task. Interestingly, prior work assumed that training with large LRs must prioritize "easy-to-fit hard-to-generalize" features [41, 68] but our results suggest that the model may favor more complex features if they are more helpful in predicting the target.

**Takeaway 3** *Using initial LRs from subregime 2A results in sparse feature learning. This effect allows for the model to focus on the most relevant features for the task (e.g., mid-frequencies in image classification), resulting in better generalization. Pre-training with other initial LR values does not show such model specialization.*

# 6 Practical setting

In this section, we demonstrate that our main results can be transferred to a more conventional training setting with some nuances. We train a common ResNet-18 model without the sphere constraint using SGD with momentum, weight decay, and data augmentation; the only deviation from the standard setup is a different LR schedule. Due to unstable behavior of non-scale-invariant weights with large LRs [35], we can only observe regimes 1 and 2. The boundary between the regimes is also less clear, mainly due to the presence of augmentations, which make the data harder to learn. Therefore, the model is unable to converge completely with small LRs, so we have drawn the boundary between the regimes approximately according to the PLR with maximum quality at the pre-trained point.

As can be seen in Figure 8, left, our first claim that the best quality is achieved with LR drops at the beginning of the second regime (subregime 2A) is confirmed. It is also clear that fine-tuning in subregime 2A converges to similar optima regardless of FLR, while at larger PLRs fine-tuning leads to diverse minima. In the first regime, the behavior is substantially different due to the mentioned issues with convergence: we see that even fine-tuning with small FLRs can still improve quality. We

note that at the best points of subregime 2A, fine-tuning reaches a test accuracy of $\sim 95\%$, which is no worse than that of a standard LR schedule [49]. Linear connectivity (Figure 8, middle) is preserved for both regime 1 and subregime 2A and lost in subregime 2B. The small train error barriers at the beginning of the first regime are due to the catapulting effect of large FLRs. Finally, frequency bands accuracies in the right plot of Figure 8 depict a similar trend as described in Section 5: in subregime 2A, the network captures significantly more mid-frequencies from the inputs than other components, while no similar specification is observed for other initial LR ranges.

In Appendix I, we provide additional practical results including analysis of SWA, angular distances, test error barriers, etc., as well as ablations on CIFAR-100 and Tiny ImageNet. We also show how our findings transfer to Vision Transformers (ViT) in Appendix J. To summarize, our main claims remain valid in the practical setting.

**Takeaway 4** *Most of the results obtained in a controlled setting are relevant to practice.*

## 7 Discussion and future research directions

**Convergence threshold**   One of the main claims of this work is that the best initial LR values lie just above the *convergence threshold* (CT). But what is this threshold and how can it be found and used in practice?

In general, by the convergence threshold for a given model and training setup we mean a learning rate value that separates regimes 1 and 2. In other words, training with a constant LR below the CT leads to convergence to a minimum, while taking a larger constant LR prevents the optimization from converging. Convergence here is defined in a conventional sense, i.e., the optimized functional value (training loss) closely approaches its global minimum by the end of training; in a simplified training setup without advanced data augmentations, it can be tracked by the ability of the model to fit the training data (i.e., reach $\sim 100\%$ training accuracy). Yet, as we show in Appendix C, CT is better understood as a small zone within the overall LR range, since the exact threshold may slightly shift depending, e.g., on the epoch budget.

That said, regimes 1 and 2 can be clearly distinguished by the behavior of the training loss in the first epochs: whether it decreases to low values or gets stuck at some non-zero level (refer to Fig. 1 in Kodryan et al. [35]). Thus, finding the CT could be done relatively efficiently using, e.g., binary search by LRs. However, in practice, precisely determining the CT is not necessary. Even though optimal initial LRs for fine-tuning with a constant small LR or weight averaging (subregime 2A) are located just above the CT, larger LRs from subregime 2B may lead to similar final quality if a more advanced LR schedule is used (see Appendix D): decreasing LR in several steps can correct for an initial value that is too high. Accordingly, most practical LR schedules are designed in that manner: starting with a large LR and gradually decreasing it as training progresses. Therefore, given a proper schedule, it is recommended to start training with a reasonably high LR value from regime 2, i.e., not allowing for convergence but also not leading to a training failure, to achieve a good final solution.

**Loss landscape and feature learning**   We have shown that the best LRs for pre-training simultaneously identify a basin with good solutions and sparsify the learned features, focusing on the most useful ones. A very intriguing question is how exactly are these observations related? Based on the results concerning subregime 2A, it can be assumed that learning some subset of features corresponds to localizing a certain region in the loss landscape, all solutions within which rely to a greater extent on these learned features. In this regard, what features were learned during the pre-training stage can determine the quality of the localized basin of solutions. This conjecture gives rise to a number of very nontrivial but interesting questions. For instance, can we somehow connect the properties of the learned features with certain characteristics of the basin and minima within it, e.g., some notion of *sharpness*? According to Kodryan et al. [35], the solutions obtained with higher PLRs of the first regime have both better generalization and lower sharpness, however this trend is more complex for the fine-tuned solutions in regime 2 (see Appendix K). Next, how exactly does feature sparsity affect the properties of the found basin: does it only pre-define a specific set of shared features or can it act as some sort of a regularizer for the solutions inside the basin, e.g., by leading to simpler models [6, 12, 55], which in fact can be represented by smaller networks? The study of the described issues opens an important direction for future research of neural networks training, as it may draw links between the optimization process and the final model properties.

**Practical implications** Despite a relatively small scale of our experiments, we provide several important practical implications and possible explanations of generally accepted practices. Firstly, we confirm that choosing higher LRs not only speeds up training, but also allows the neural network to find solutions that are inaccessible at low LRs, which was previously shown only in specific settings [41, 44]. Secondly, we discover that it is important both to choose the initial LR and to design a full LR schedule. A good choice of the initial LR allows for localizing a region with the best solutions, as too small LRs tend to converge to the nearest optima with poor generalization. At the same time, designing a suitable LR schedule is necessary if training is started with a too high LR: in that case, it is important not to decrease it too quickly, but to gradually go through the optimal values in order to be able to localize a good basin before convergence.

We also show that the choice of the initial LR can lead to different feature learning behavior. In our setting, sparse features and mid-frequency bias were associated with optimal generalization. However, in practice this may not always be the case. For example, some features useful for the training data classification can be spurious for the test data [33]. Or memorization, in general affecting a very small subset of data, is less likely to happen when training with large LRs, while some works show that memorization in neural networks can be useful [18]. Thus, the properties of training with different LRs in more complex practical scenarios with spurious features/benign memorization may lead to more complex relationships between LRs and generalization. We believe this direction is significant for future research to more broadly understand the impact of learning rate on trained models.

# 8    Conclusion

In this work, we studied the influence of training with different initial LRs on the properties of the final solution. We discovered that pre-training with moderately large LRs, slightly above the convergence threshold, provides the best points for subsequent fine-tuning or weight averaging. From the geometry perspective, training with these LR values locates a basin of well-generalizing solutions in the loss landscape; from the feature learning perspective, these solutions correspond to a sparse set of learned features that are most useful for the task. Using other LR values may lead to suboptimal results: either unstable local minima corresponding to a dense set of learned features with smaller LRs or vast areas with diverse minima and degraded feature learning with larger LRs. We conduct main experiments in a special setup allowing for more accurate control of the learning rate and validate our key results in a practical setting. Our findings can be useful for both practical and theoretical future work on optimizing LR schedules, loss landscape structure, and feature learning in neural networks.

**Limitations** We wish to highlight several limitations of our work. First, our study is primarily empirical in nature; our conclusions do not have direct theoretical support (perhaps only indirect via related work partly mentioned in Section 1.1). Second, we are limited to a specific setup involving particular datasets (image or synthetic) and NN architectures (convolutional, MLP, or Vision Transformer) and, generally speaking, cannot guarantee that all of our findings will consistently generalize to other settings. Third, although we account for the impact of scale invariance on LR in our main experiments, we may overlook similar effects of other NN invariances, like rescale invariance of homogeneous activations (e.g., ReLU) [21, 45]. Addressing these and other possible limitations is future work.

# Acknowledgments

The related work analysis in Section 1.1 and the overall text composition were done by Maxim Kodryan with the support of the grant for research centers in the field of AI provided by the Analytical Center for the Government of the Russian Federation (ACRF) in accordance with the agreement on the provision of subsidies (identifier of the agreement 000000D730321P5Q0002) and the agreement with HSE University №70-2021-00139. The empirical results were supported in part through the computational resources of HPC facilities at HSE University [36]. Part of the experiments were conducted using the Constructor Research Platform [14].

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

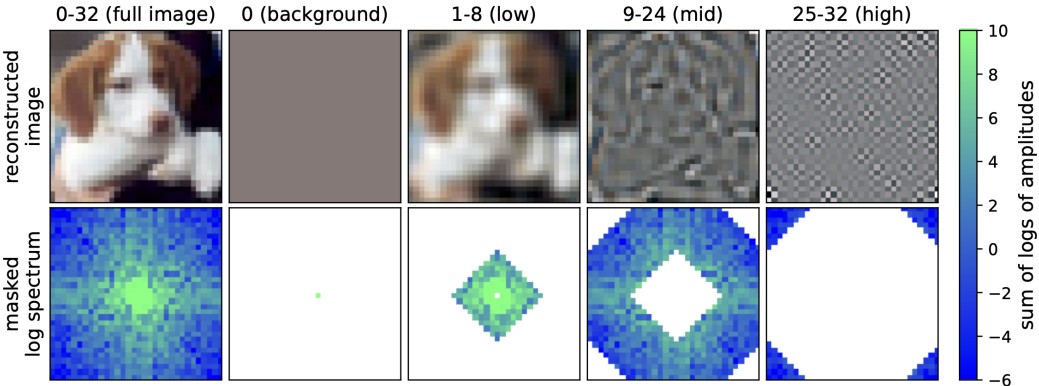

Figure 9: Inverse 2D DFT images (top) and corresponding masked spectra (bottom). When visualizing the *low*, *mid*, and *high* images, we scale each channel to the range 0-1. For the spectra, we plot the logarithm of the absolute values of the amplitudes ($\log |Y[k, l]|$), summed over 3 color channels.

## A   Experimental setup details

**Code**   The code for our main experiments can be found in the following repository: `https://github.com/isadrtdinov/understanding-large-lrs`. Our implementation, including the used scale-invariant architectures and training on the sphere, is based on the open-source code of Kodryan et al. [35]: `https://github.com/tipt0p/three_regimes_on_the_sphere`.

**Compute resources**   We use NVIDIA TESLA V100 and A100 GPUs for computations in our experiments. The total amount of compute spent on all experiments is approximately 1500-2000 GPU hours, while the experiments included in the paper took $\sim$ 1000 GPU hours.

**Datasets and architectures**   Following Kodryan et al. [35], we conduct most of our experiments with two network architectures, a simple 3-layer convolutional neural network with Batch Normalization layers [24] (ConvNet) and a ResNet-18, on CIFAR-10 and CIFAR-100 datasets. In the scale-invariant setup, we use ResNet models with width factor $k = 32$, and ConvNet models with width factor $k = 32$ and 128 for CIFAR-10 and CIFAR-100, respectively. In the practical setup, we use ResNet with a standard width factor $k = 64$ and additionally consider the Tiny ImageNet dataset and Vision Transformer (ViT) architecture.

**Pre-training, fine-tuning, and SWA**   We train all networks using SGD with a batch size of 128. Both the pre-training and the fine-tuning stages take 200 epochs. This training time is sufficient to either reach a minimum or stabilize the loss in the pre-training stage and to achieve complete convergence in the fine-tuning stage even with the smallest FLR. Fine-tuning is always done with the first regime FLRs to ensure convergence to a minimum. When performing SWA of $N$ models, we continue training for $N - 1$ more epochs with the same PLR and average checkpoints from epochs $200, \ldots, 200 + N - 1$. In the practical setting in Section 6, we use weight decay of $5 \cdot 10^{-4}$, momentum of 0.9, and standard augmentations: random crops (size: 32 for CIFAR and 64 for Tiny ImageNet, padding: 4), random horizontal flips, and per-channel normalization.

**Synthetic example**   We use a 3-layer MLP with ReLU activation and Layer Normalization [8] after the first and the second linear layers to make the network scale-invariant. Additionally, we freeze the final linear layer and set its norm to 10. The size of hidden layers is 32. The trainable weights are initialized with the standard normal distribution and projected to the unit sphere. We take 512 training and 2000 testing samples. We use SGD with batch size 32. Pre-training and fine-tuning stages take 40000 and 20000 iterations, respectively. We consider 10 data sampling seeds and 5 model initialization + SGD batch order seeds, so a total of 50 training runs is done.

**Fourier features** We use 2D Discrete Fourier Transform (DFT) and frequency masking to create images similar to the single-feature samples in the synthetic example. Let $X \in \mathbb{R}^{N \times M}$ denote an image in the spatial domain (for CIFAR-10/CIFAR-100 we have $N = M = 32$; for Tiny ImageNet we have $N = M = 64$). 2D DFT is defined as:

$$Y[k,l] = \frac{1}{NM} \sum_{n=0}^{N-1} \sum_{m=0}^{M-1} X[n,m] e^{-2\pi i (\frac{kn}{N} + \frac{lm}{M})}, \quad \begin{cases} -\lfloor \frac{N}{2} \rfloor \leq k \leq \lceil \frac{N}{2} \rceil - 1 \\ -\lfloor \frac{M}{2} \rfloor \leq l \leq \lceil \frac{M}{2} \rceil - 1 \end{cases}$$

In the case of CIFAR images, e.g., $-16 \leq k, l \leq 15$. The frequency band $b$ is defined as $\{(k,l) : |k| + |l| = b\}$, corresponding to a diamond around the spectrum center $k = l = 0$. This band matches the Fourier basis vectors, which have $b$ oscillations ($b$ "black and white" stripes) rotated at different angles. Then, we apply frequency masking, preserving a range of frequency bands, $a \leq b \leq c$:

$$Y_{a\text{-}c}[k,l] = \begin{cases} Y[k,l], & a \leq |k| + |l| \leq c \\ 0, & \text{otherwise} \end{cases}$$

Finally, we use the inverse 2D DFT and obtain the resulting frequency band images:

$$X_{a\text{-}c}[m,n] = \sum_{k=-\lfloor N/2 \rfloor}^{\lceil N/2 \rceil - 1} \sum_{l=-\lfloor M/2 \rfloor}^{\lceil M/2 \rceil - 1} Y_{a\text{-}c}[k,l] e^{2\pi i (\frac{kn}{N} + \frac{lm}{M})}$$

We use 0-0 (constant background color), 1-8 (low), 9-24 (mid), and 25-32 (high)[5] groups of frequency bands. Each of the RGB color channels is processed independently. An example of the application of the described procedure is shown in Figure 9. We omit the per-channel image normalization used during training when evaluating accuracy on low, mid, and high samples (since removing the 0 band centers the resulting images). However, it is still used for the 0-0 sample.

## B Generalization analysis for other datasets and architectures

In Figure 10, we demonstrate the test accuracy after pre-training with different PLR values (black line) and after fine-tuning with given FLRs or SWA (colored lines) for other dataset-architecture pairs: SI ConvNet on CIFAR-10/CIFAR-100 and SI ResNet-18 on CIFAR-100. The results are similar to that of SI ResNet-18 on CIFAR-10, described in the main text and shown in Figure 2.

We again can divide each plot into three parts w.r.t. the three pre-training regimes, and the main takeaways also hold. Pre-training test accuracy is monotonic in the first regime and both fine-tuning and SWA are unable to improve it for high PLRs of the first regime. Optimal PLRs for further fine-tuning/SWA are attributed to the beginning of the second regime, while the quality deteriorates by the end of the second regime. Fine-tuning with low FLRs in the third regime is better than training with the corresponding LR from scratch starting at the standard random initialization.

There are, however, two remarks. First, for the ConvNet model the effects of the second regime are less pronounced: both fine-tuning and SWA show less improvement in subregime 2A and the deteriorating effect of subregime 2B on fine-tuning is almost leveled out. We suppose that it could be explained by the simplicity of ConvNet, which allows more robust training with a fixed LR and much less scope for further quality improvement. Second, due to the periodic behavior [43] of ResNet-18 trained on CIFAR-100 with high first regime LRs, also reported by Kodryan et al. [35], we observe instabilities for SWA of more than 10 models, since checkpoints from different periods lay in different low-loss regions in that case.

## C Boundary between the first and second regimes

In this section, we motivate our statement in the main text that the best quality obtained after fine-tuning from the second regime is unattainable by training from scratch with a fixed LR for any reasonable time. For that purpose, we train our models till convergence (with a maximum of $2 \cdot 10^4$ epochs), i.e., when training loss value reaches $10^{-3}$, and track the convergence time and the achieved test accuracy for different LRs near the boundary between regimes 1 and 2 (see Figure 11).

---

[5]This range is for CIFAR-10/CIFAR-100. For Tiny ImageNet, we use 25-64 bands as high frequencies.

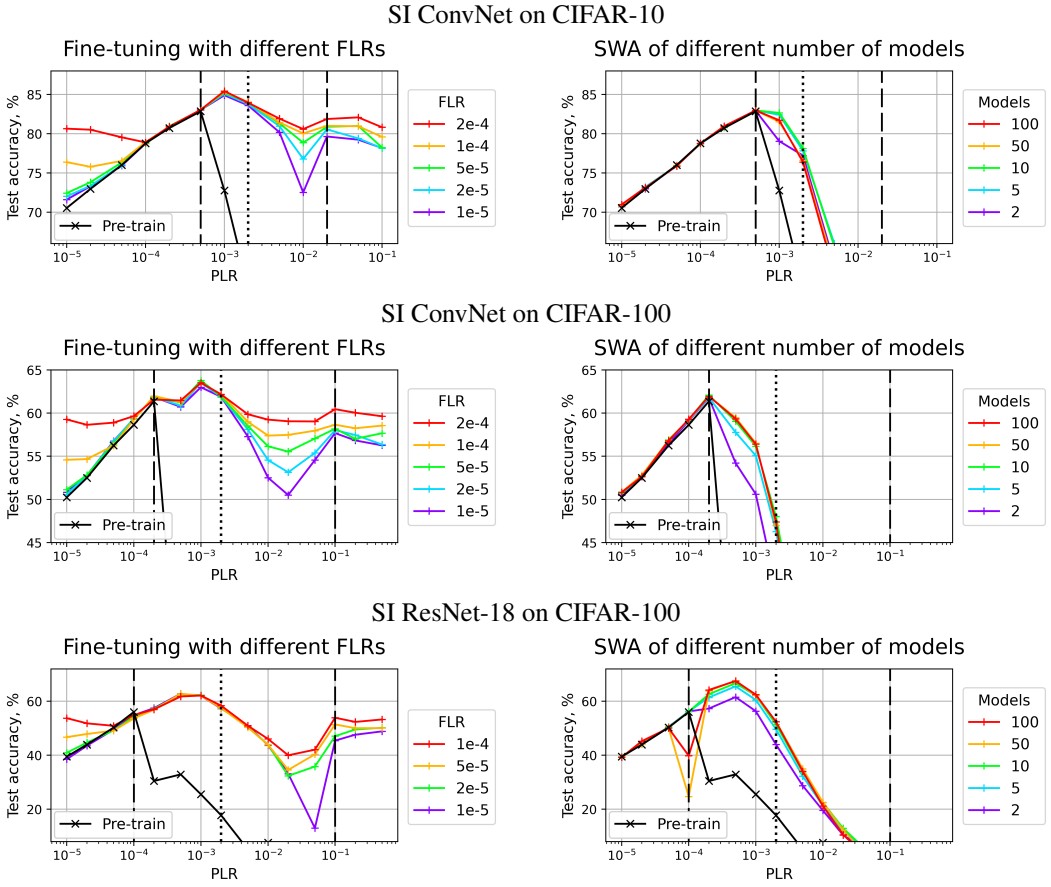

Figure 10: Test accuracy of different fine-tuned (left) and SWA (right) solutions. Test accuracy after pre-training is depicted with the black line. Dashed lines denote boundaries between the pre-training regimes, dotted line divides the second regime into two subregimes. Results for other dataset-architecture pairs, similar to Figure 2.

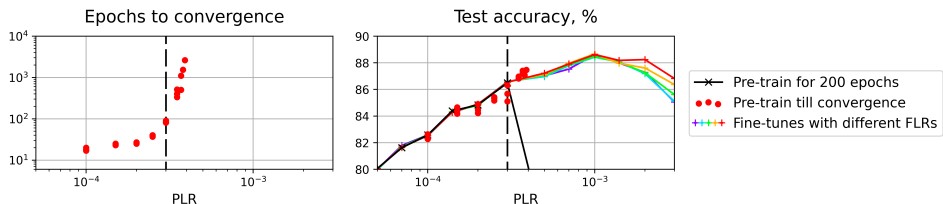

Figure 11: Number of training epochs to convergence (left) and test accuracy (right) for different PLRs on the boundary between regimes 1 and 2. Red points are obtained after training to convergence from scratch with a fixed LR value (we run each experiment with three different seeds).

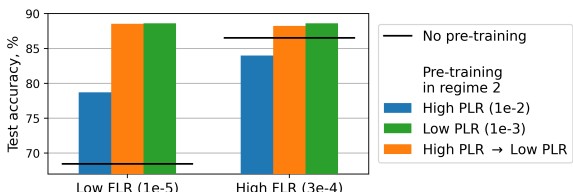

Figure 12: Test accuracy obtained after fine-tuning with two different FLR values. Blue bar denotes fine-tuning after pre-training with a PLR from subregime 2B, green bar denotes fine-tuning after pre-training with a PLR from subregime 2A, and orange bar denotes first fine-tuning with a PLR from subregime 2A and then with a given FLR after pre-training with a PLR from subregime 2B. Black lines denote training from scratch with a given FLR.

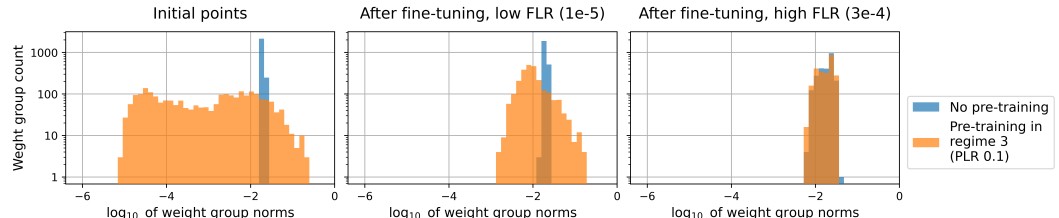

Figure 13: Histograms of individual scale-invariant weight group norms for standard random initialization (blue) and pre-training with a third regime PLR (orange). Left plot shows norms right after initialization/pre-training, middle plot shows norms after fine-tuning with a low FLR, right plot shows norms after fine-tuning with a high FLR.

We observe that the required training time increases sharply after the threshold separating the regimes for a 200 epochs budget. The obtained minima have approximately the same test accuracy as the fine-tuned solutions at the corresponding PLRs, however, they take immensely more epochs to reach. Based on the training time growth, we hypothesize that the best fine-tuning test accuracy obtained with PLR $\approx 10^{-3}$ is unattainable with any realistic epoch budget. Therefore, we conclude that training from scratch with a fixed LR from a standard initialization does not allow to find as good optima as are available after pre-training in subregime 2A.

## D    Gradual fine-tuning restores the quality for higher second regime PLRs

In this section, we provide additional results on fine-tuning after pre-training in subregime 2B. As stated in the main text, in that case, immediate LR drop to the first regime leads to suboptimal fine-tuning quality compared with subregime 2A. However, we found that adding only one additional step in the LR schedule improves the fine-tuning test accuracy almost to the optimal level. Specifically, after pre-training with a PLR of subregime 2B, we first drop LR to a lower PLR attributed to subregime 2A, fine-tune for 200 epochs, then drop LR once again to a given FLR of the first regime and fine-tune for 200 more epochs until convergence. Such a two-step LR schedule helps achieve almost the same test accuracy as usual fine-tuning with the same FLR after pre-training in subregime 2A, which gives the best solution.

Figure 12 shows test accuracies of the fine-tuned solutions obtained after pre-training with a subregime 2B PLR (blue bars), pre-training with a subregime 2A PLR (green bars), and two-step LR schedule through the subregime 2A PLR (orange bar) for the highest and the lowest FLR values. Black lines denote the respective test accuracies after training from scratch with the given FLR values. We can see that indeed gradual fine-tuning through lower second regime LRs restores the quality for higher second regime PLRs.

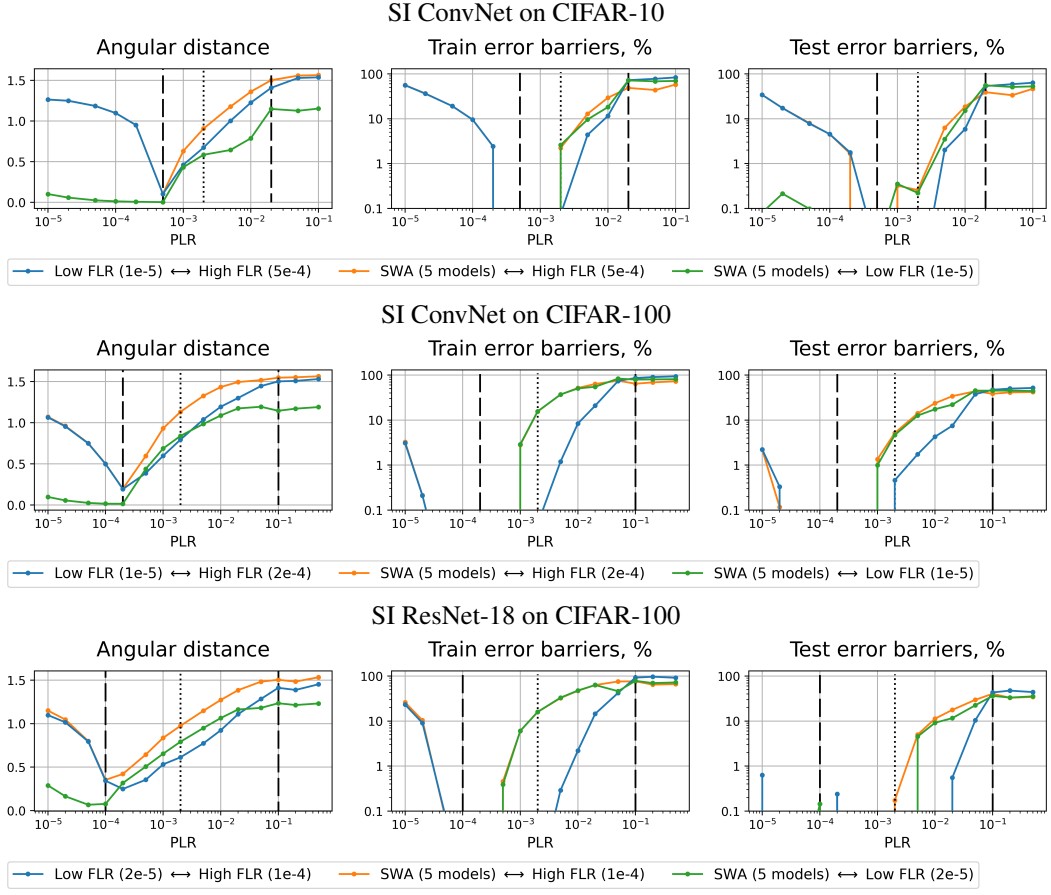

Figure 14: Geometry between the points fine-tuned with the smallest and the largest FLRs and SWA. Results for other dataset-architecture pairs, similar to Figure 3.

# E  Pre-training in regime 3: fine-tuning with low FLRs

In this section, we give more detail on reasons for better fine-tuning results with low FLRs after pre-training in the third regime compared to training from scratch with the same FLRs. When training with a fixed learning rate $\eta$, ELR for a scale-invariant parameter group $\theta$ is defined as $\eta / \|\theta\|^2$. In the main text, we suggest that a possible explanation could be the uneven distribution of norms of individual scale-invariant groups resulting in the corresponding uneven distribution of individual ELRs after pre-training with very large PLRs. That means, that when fine-tuning from the third regime, some weight groups are learning faster than the others, which gives them the benefits of large learning rate training despite the low total LR. In contrast, with a standard initialization all weight norms are approximately the same, which implies that the whole model is essentially trained with the same small effective learning rate, resulting in inferior quality.

In Figure 13, we show the histograms of norms of individual scale-invariant weight groups for a standard initialization (blue) and pre-training with a third regime PLR (orange). We depict three stages: right after initialization/pre-training (left), after training from scratch/fine-tuning with a low FLR (middle), and after training from scratch/fine-tuning with a high FLR (right). We see that the weight norms distribution is more spread after pre-training. Remarkably, this effect persists after fine-tuning with a low FLR and is almost eliminated after fine-tuning with a high FLR, which does not have any advantages over training from scratch.

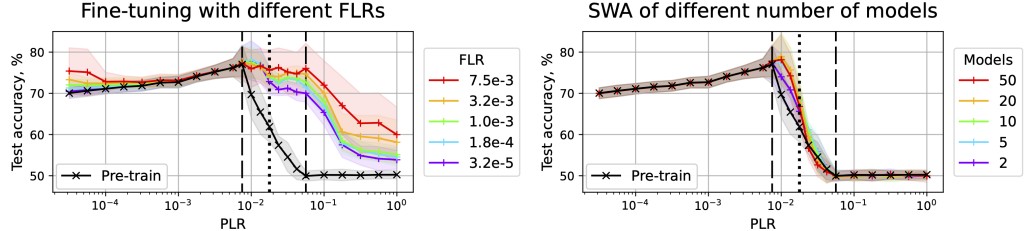

Figure 15: Syntetic example. Test accuracy of different fine-tuned (left) and SWA (right) solutions. Test accuracy after pre-training is depicted with the black line. Dashed lines denote boundaries between the first and second pre-training regimes, dotted line divides the second regime into two subregimes. Mean and standard deviation over 50 seeds are shown.

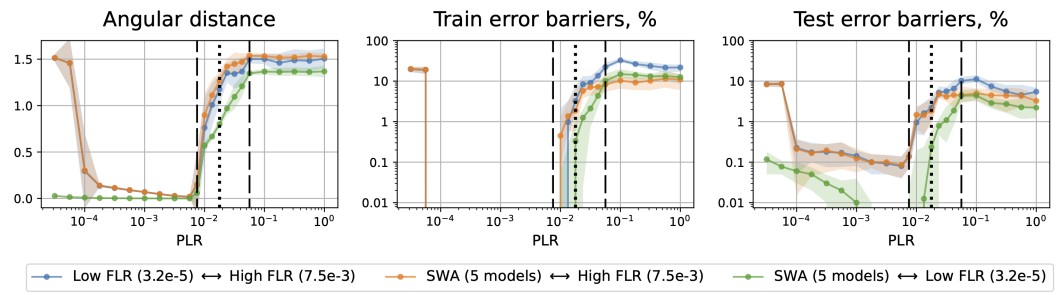

Figure 16: Geometry in the synthetic example between the points fine-tuned with the smallest and the largest FLRs and SWA. Results are aggregated over 50 seeds. For angular distance, mean and standard deviation are shown; for train and test barriers, median and $0.25 - 0.75$ quantile range are plotted.

## F Loss landscape analysis for other datasets and architectures

In this section, we ablate the geometric properties of pre-training in different regimes for other dataset-architecture pairs. Figure 14 exhibits the full panel of results.

As can be seen, as with the generalization ablations in Appendix B, our main claims hold for other architectures and datasets. In the first regime, high FLRs lead to catapults and convergence to a new separate minimum, while small FLRs and SWA end up in effectively the same mode. In subregime 2A, the distances between the fine-tuned and SWA points remain small with almost no barriers between them. However, we note that closer to the right end of subregime 2A linear connectivity with the SWA solution can be lost for more complex settings such as training on CIFAR-100. This behavior is expected as the SWA point is obtained after averaging subsequent pre-trained checkpoints, which can be located at a considerable distance from each other due to relatively large PLRs, unlike the fine-tuned solutions, which come from the same pre-trained checkpoint. This is also reflected in higher angular distance between SWA and each of the fine-tuned points than between the fine-tuned points. Finally, after subregime 2A the obtained points lose linear connectivity and lie further apart in angular distance.

## G Additional result for the synthetic example

In this section, we provide additional results for the synthetic example. Figure 15 shows the test accuracy of the fine-tuning with different FLRs and SWA. The overall behavior for different PLRs is similar to the main experimental setup, all 3 regimes are clearly observed. In regime 1, the pre-training quality is monotonically increased with PLR. Subsequent fine-tuning with smaller, equal, or slightly larger FLRs leads to the same quality, while a significantly larger FLR improves test accuracy. Fine-tuning in subregime 2A gives a slight improvement over training models in regime 1, and all FLRs have the same optimal quality (except for higher FLRs, which experience overfitting given a fixed number of fine-tuning iterations). Fine-tuning with different FLRs in subregime 2B leads to

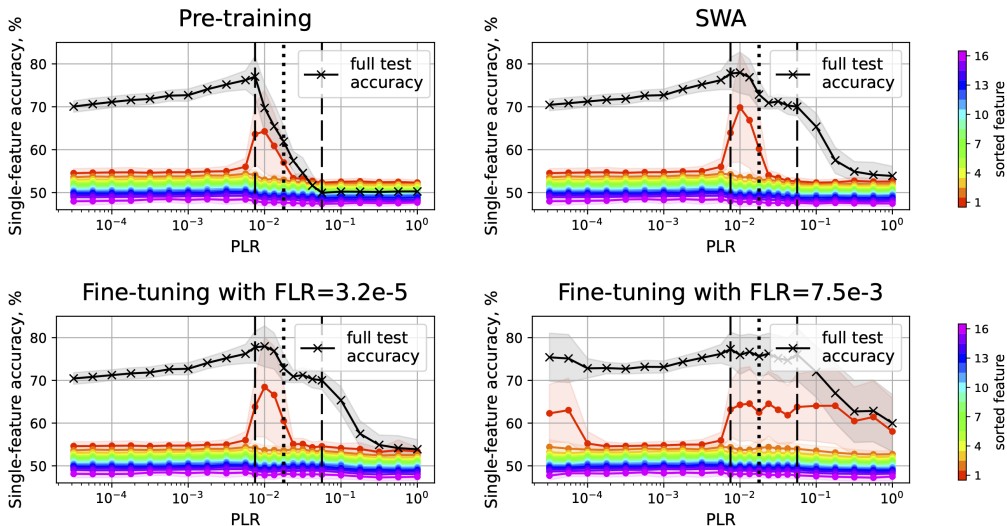

Figure 17: Feature sparsification in the synthetic example for pre-training (top, left), SWA (over 5 models; top, right), and fine-tuning with low FLR $= 3.2 \cdot 10^{-5}$ (bottom, left) and high FLR $= 7.5 \cdot 10^{-5}$ (bottom, right). Mean and standard deviation over 50 seeds are shown.

solutions with different accuracy, which is lower compared to 2A. Finally, SWA in subregime 2A produces models with good performance, while averaging models in subregime 2B does not.

Figure 16 shows angular distances and error barriers for the synthetic example. Similarly to the main setup, we observe the catapult effect for FLR $\gg$ PLR. Although both train and test error barriers emerge in subregime 2A, the barrier values are significantly higher in subregime 2B. It is noteworthy that the angular distance and the error barriers between SWA and low FLR are much smaller than those to the high FLR (at least up to the boundary between subregime 2B and regime 3). Overall, most claims from the Takeaway 2 hold.

Lastly, Figure 17 complements Figure 5 from the main text. We observe that SWA and fine-tuning with a low FLR preserve feature sparsity obtained from pre-training in subregime 2A. Pre-traning with other initial LR values does not allow the model to focus on a single feature. However, fine-tuning with a high FLR restores feature sparsity by either the catapults (when FLR $\gg$ PLR, regime 1) or convergence (when FLR $<$ PLR, regimes 2 and 3).

## H  Additional results on Fourier features

In Figure 18 we present the accuracy of different frequency bands for the rest of scale-invariant setups. Considering SI ResNet-18 on both datasets, the mid-frequency features have higher absolute accuracy values in subregime 2A compared to background and low-frequency features (both in regimes 1 and 2). The specialization on mid frequencies is even more pronounced after weight averaging and fine-tuning. Moreover, the behavior of mid-frequency line after fine-tuning is highly correlated with the test accuracy of the corresponding FLRs in Figures 2,10.

As for the SI ConvNet architecture, feature specialization is less obvious: we do not observe significant focus on mid frequencies in subregime 2A, perhaps because a combination of 3 convolutional layers is not enough to learn fine-grained image details. This lack of sparsity may be also related to less pronounced effects of the second regime, discussed in Appendix B. Nevertheless, we see a peak of mid-frequency accuracy in subregime 2A for both CIFAR-10 and CIFAR-100, indicating that pre-training with larger PLRs is beneficial for this setup too.

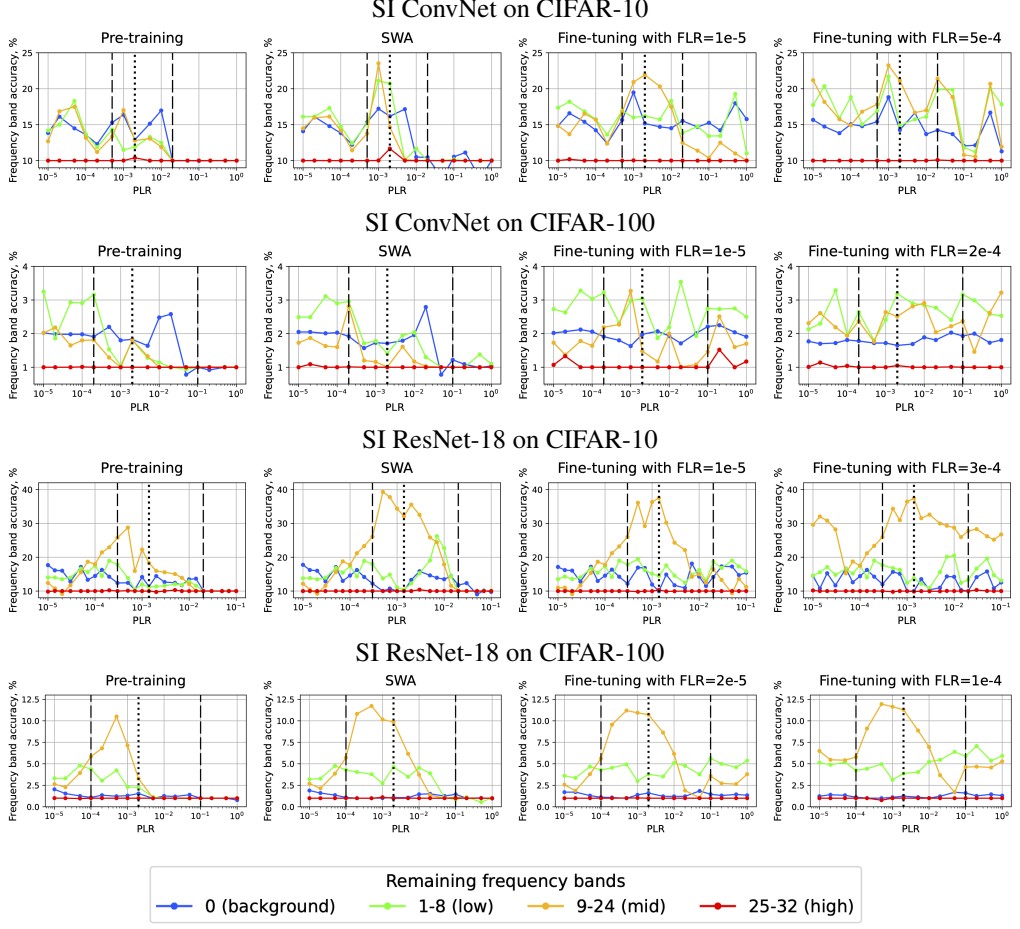

Figure 18: Accuracy of different frequency bands for pre-training (column 1), SWA (over 5 models; column 2), and fine-tuning with low FLR (column 3) and high FLR (column 4). SI ConvNet and SI ResNet-18 on CIFAR-10/CIFAR-100.

# I  Additional results for the practical setup

In this section, we provide additional results for the practical setup of our experiments: additional plots for the ResNet-18 on CIFAR-10 and ablation on the CIFAR-100 and Tiny ImageNet datasets.

Figure 19 depicts the generalization results for the fine-tuned (left plots) and SWA (right plots) solutions in the practical setup. SWA follows the same trend as fine-tuning, confirming our main claim: the best solutions are achieved in the lower part of the second regime, i.e., subregime 2A. At larger PLRs of the second regime SWA quality rapidly deteriorates. In the first regime, both SWA and fine-tuning are able to improve the accuracy due to incomplete convergence discussed in the main text. The results on all datasets are similar. Small accuracy drops for pre-training with $\mathrm{PLR} = 10^{-3}$ on Tiny ImageNet and fine-tuning with $\mathrm{FLR} = 2 \cdot 10^{-3}$ from $\mathrm{PLR} = 10^{-3}$ on CIFAR-100 are due to the periodic behavior [43]: the final epoch occurs at the beginning of a period. Similar effects are discussed above, at the end of Appendix B, when training SI ResNet-18 on CIFAR-100. That could be fixed by choosing a different random seed and/or epoch budget.

In Figure 20, we provide the geometrical results for the practical setting. The results are again similar for all datasets. We measure angular distance in this setup for consistency with the previous results. However, since we consider only scale-invariant parameters when measuring the angular distance, its value does not reflect the total distance w.r.t. all the parameters of the model.[6] Despite this fact, it still covers the majority of model parameters and behaves similarly to the controlled setting: the

---

[6]Note that $L_2$ distance is also not the best choice due to scale invariance.

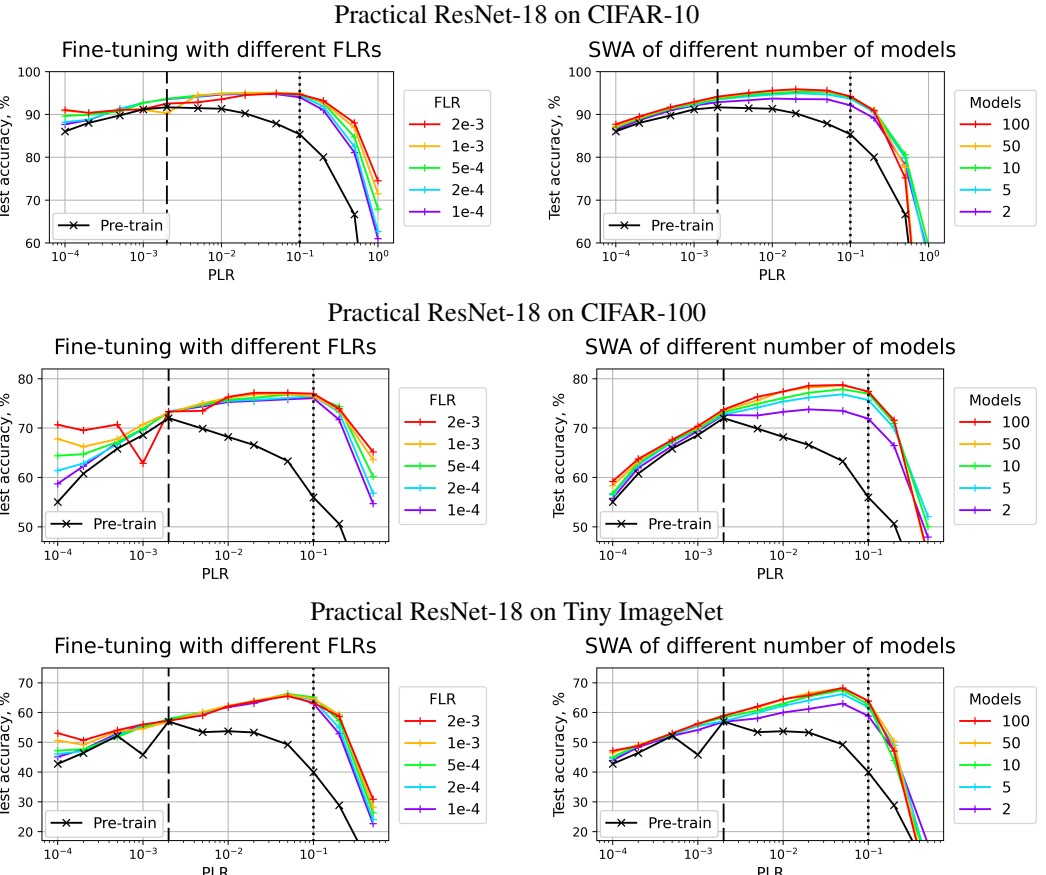

Figure 19: Practical setting. Test accuracy of different fine-tuned (left) and SWA (right) solutions. Test accuracy after pre-training is depicted with the black line. Dashed lines denote boundaries between the pre-training regimes, dotted line divides the second regime into two subregimes. Full version of the results for CIFAR-10 complementing Figure 8 and similar results for CIFAR-100 and Tiny ImageNet.

distances are high between different FLRs in the first regime, decrease in subregime 2A and then grow again. Both train and test error barriers follow the same trend as the angular distance.

Finally, Figure 21 shows the full panel of results on frequency bands analysis for the practical setting. The mid-frequency bias in subregime 2A and the induced feature sparsity is especially pronounced after fine-tuning or SWA, however it still can be clearly seen already after the pre-training stage on CIFAR-100 and Tiny ImageNet. Other PLR ranges show no such bias towards a particular spectrum component, however, fine-tuning with a high FLR can help introduce it. Overall, the mid-frequency bias is more consistent on more complex datasets, indicating a higher usefulness of this band for image classification.

## J   Additional results for Vision Transformer

In this section, we show that our main findings can be transferred to the transformer architecture. Specifically, we use a ViT small architecture [16] with patch size 4 and hidden size 512 (ViT-S/4) for $32 \times 32$ images of the CIFAR-10/100 datasets.[7] We additionally insert a LayerNorm [8] after each linear layer in the feed-forward networks. This modification increases the number of scale-invariant parameters of the transformer and makes the second regime more stable. We pre-train and fine-tune both for 500 epochs with the same protocol as in the main text and use the RandAugment(2, 14)

---

[7]We use an open source implementation by Kentaro Yoshioka [67].

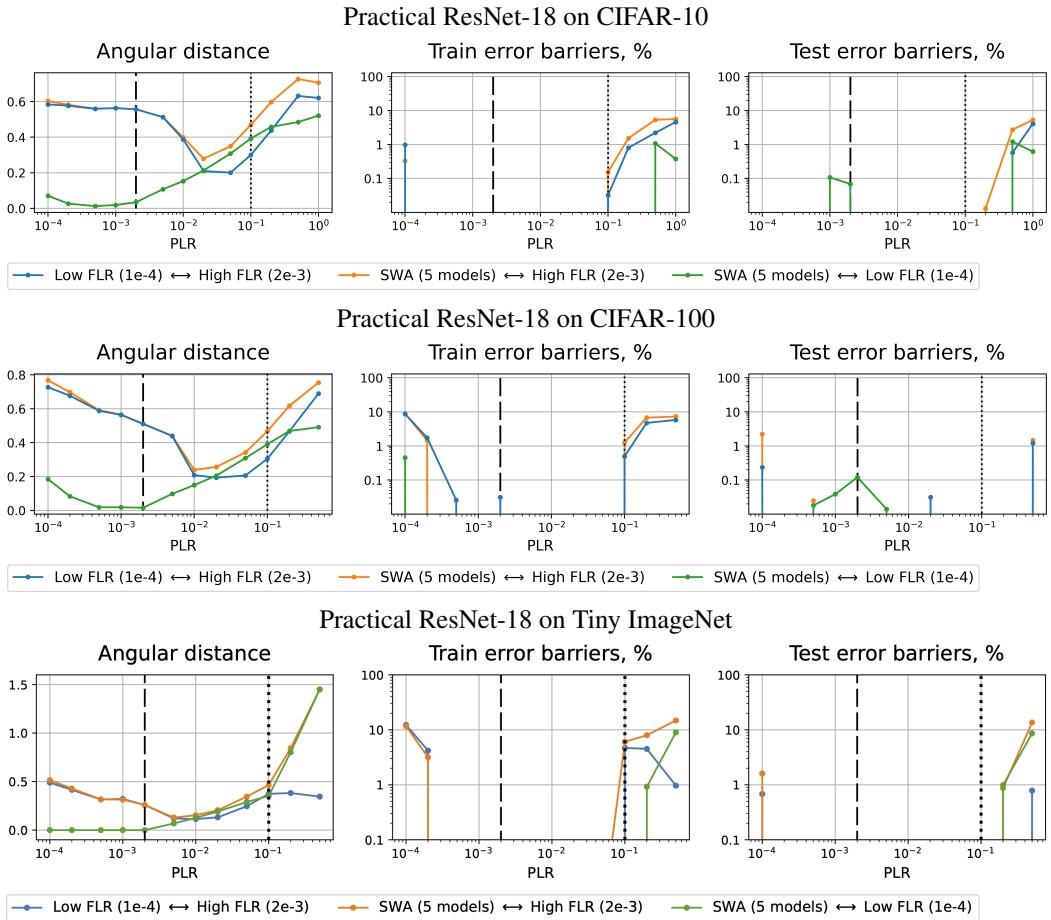

Figure 20: Practical setting. Geometry between the points fine-tuned with the smallest and the largest FLRs and SWA. Full version of the results for CIFAR-10 complementing Figure 8 and similar results for CIFAR-100 and Tiny ImageNet.

augmentation strategy [15]. We use the Adam [32] optimizer with weight decay $10^{-4}$, batch size $512$, and disabled AMP [47].

Figure 22 shows the test accuracy of fine-tuned (left plots) and SWA (right plots) solutions of ViT-S/4 on both datasets. At the same time, Figure 23 depicts the angular distance and the train/test barriers between solutions fine-tuned with different FLRs and SWA. The general trends are the same as for the practical ResNet-18, although the advantage of pre-training in subregime 2A is slightly less obvious.

The accuracy of different frequency bands for this setup is presented in Figure 24. Notably, feature learning in transformers is different from convolutional networks, since they tend to inherently capture lower frequencies in the data [54]. We can still see that the role of the most important features (low frequencies in this case) increases towards 2A. Interestingly, however, the importance of mid-frequency features also peaks in subregime 2A for both datasets, which is consistent with our intuition that mid-frequencies are essential for natural image classification.

For the sake of ablation, we also consider different partitions into low and mid-frequencies. That allows to clarify whether ViTs actually rely more on the lower frequency part of the spectrum, or whether its mid-frequencies simply "start earlier" compared to convolutional models. Figure 25 shows that the latter appears to be the case. That is, if 5-24 or 3-24 bands are selected for the mid-frequencies, then this spectrum component will dominate the low-frequency bands in terms of the corresponding test accuracy, which is similar to convolutional models.

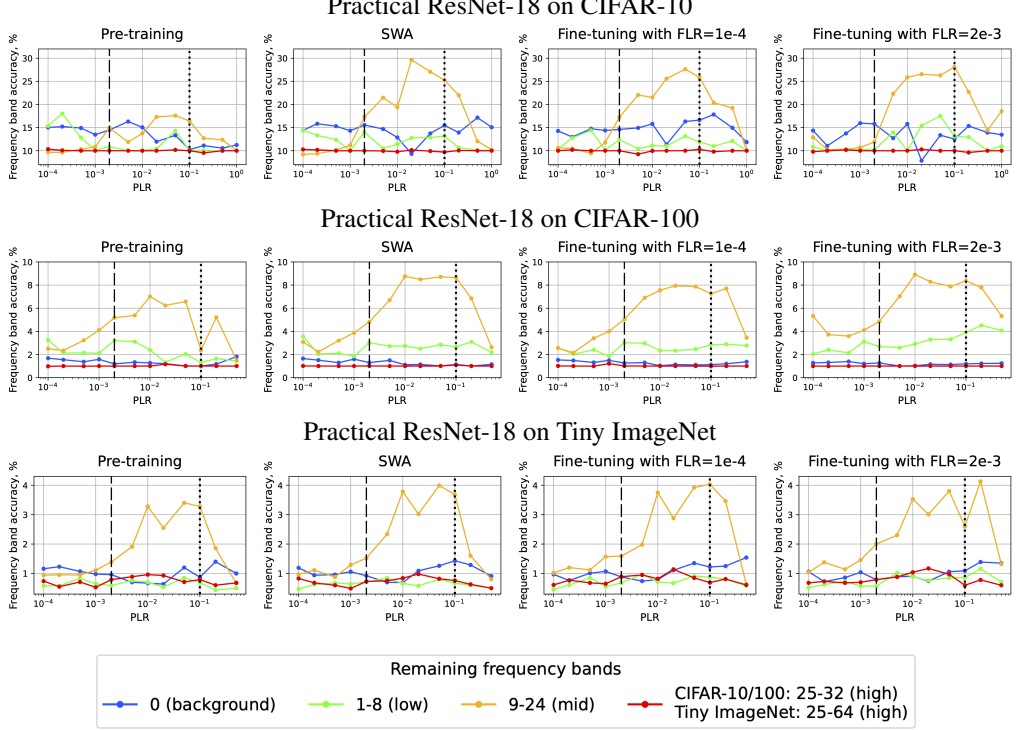

Figure 21: Practical setting. Accuracy of different frequency bands for pre-training (column 1), SWA (over 5 models; column 2), and fine-tuning with low FLR (column 3) and high FLR (column 4). Full version of the results for CIFAR-10 complementing Figure 8 and similar results for CIFAR-100 and Tiny ImageNet.

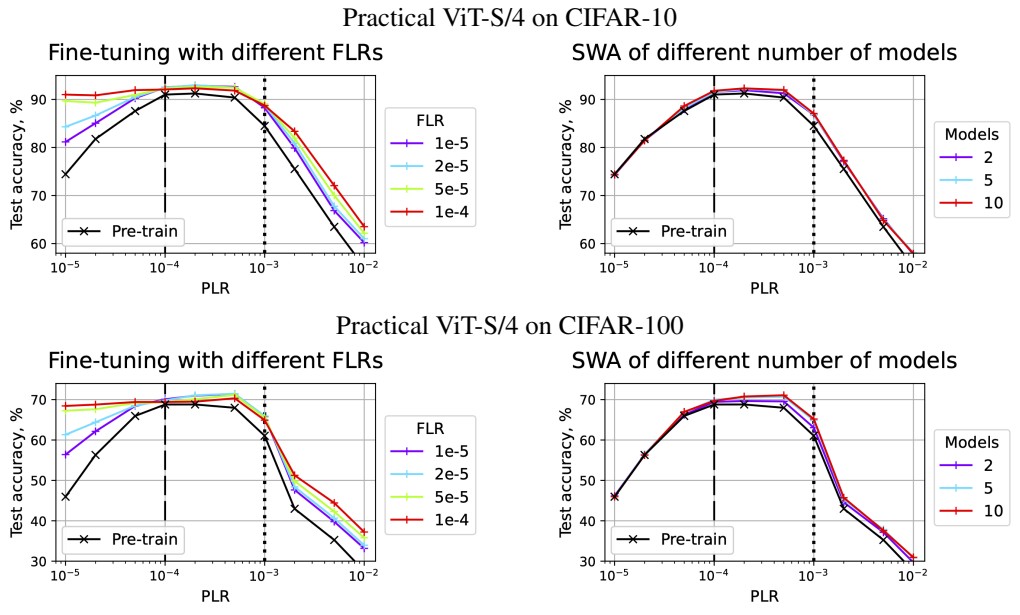

Figure 22: Practical setting on ViT. Test accuracy of different fine-tuned (left) and SWA (right) solutions. Test accuracy after pre-training is depicted with the black line. Dashed lines denote boundaries between the pre-training regimes, dotted line divides the second regime into two subregimes.

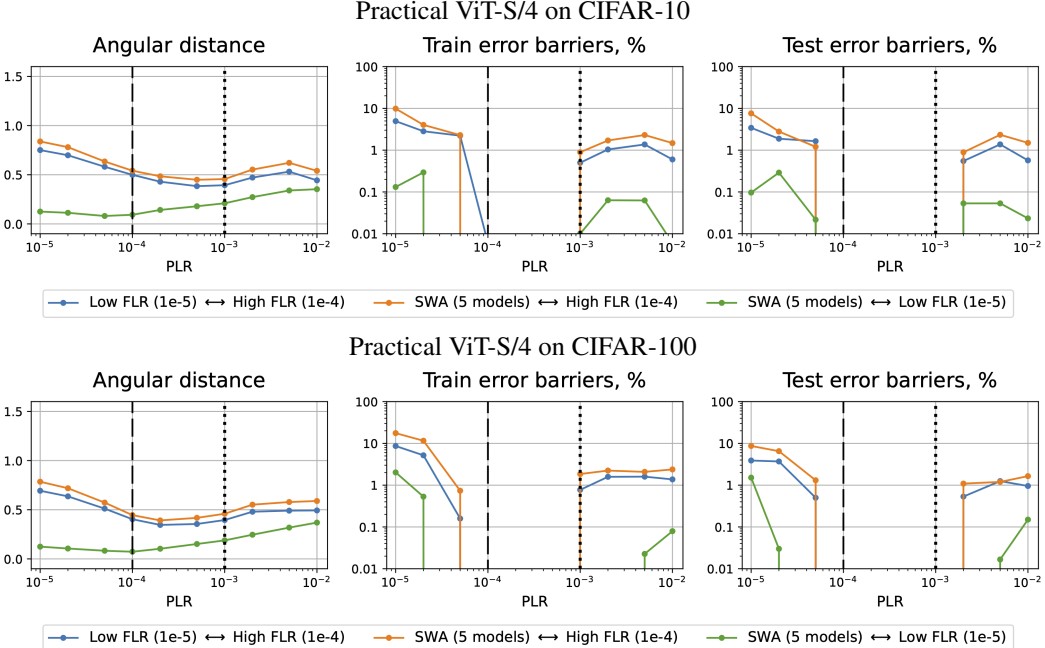

Figure 23: Practical setting on ViT. Geometry between the points fine-tuned with the smallest and the largest FLRs and SWA.

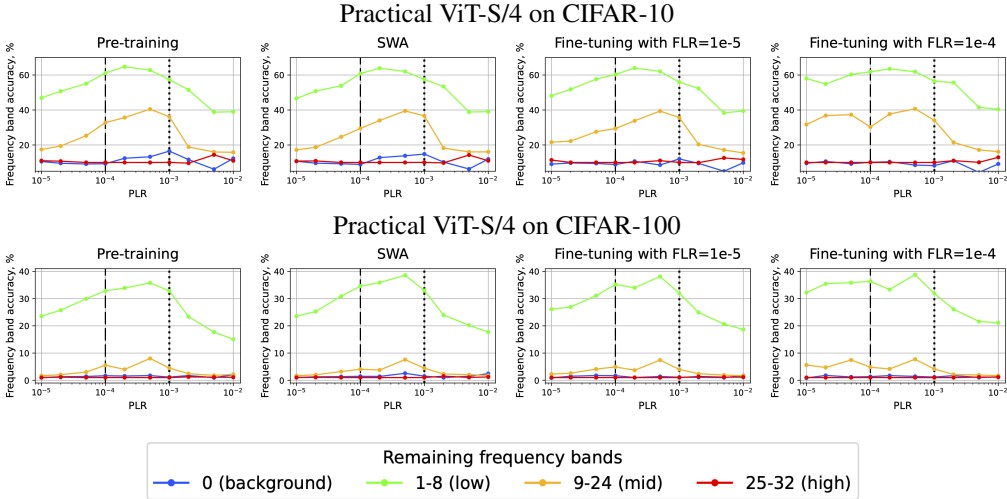

Figure 24: Practical setting on ViT. Accuracy of different frequency bands for pre-training (column 1), SWA (over 5 models; column 2), and fine-tuning with low FLR (column 3) and high FLR (column 4).

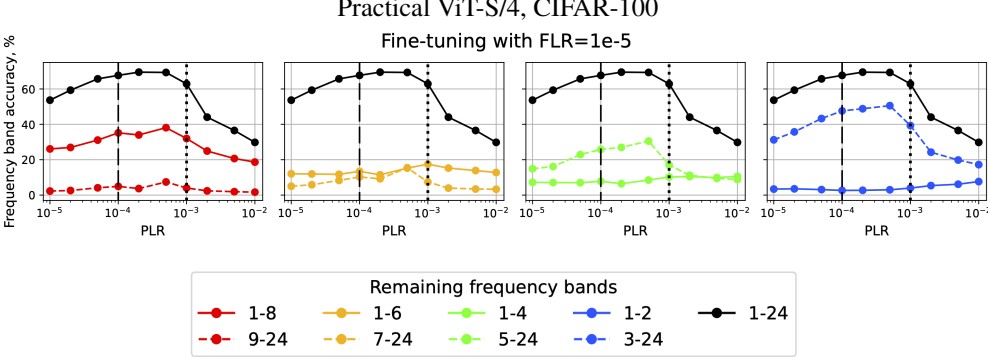

Figure 25: Practical setting on ViT. Accuracy of different frequency bands when varying the boundary between low and mid-frequencies for fine-tuning with small FLR.

## K    Generalization and sharpness of the fine-tuned solutions

In this section, we test the claim of Kodryan et al. [35] that higher LRs lead to both better generalizing and less sharp minima. Kodryan et al. [35] empirically confirmed that for the learning rates of the first regime, and we attempt to verify this result in the second regime.

To this end, we measure test error and sharpness of the fine-tuned solutions obtained after pre-training in regime 2. We adopt the measure of sharpness used by Kodryan et al. [35], which is the mean stochastic gradient norm over batches of the data. For fairness of comparison, since lower loss may naturally lead to lower gradients, we set the training loss of all the fine-tuned solutions to $6 \cdot 10^{-4}$. We do this by taking an appropriate weighted average of the test error and sharpness values measured at two consecutive checkpoints: just before and right after crossing the set loss threshold.

The results are presented in Figure 26. At first sight, the picture shows an overall positively correlated trend in the relationship between test error and sharpness. A similar trend is observed in separate groups of fine-tuned solutions obtained from the same initialization (i.e., pre-trained checkpoint). However, by comparing fine-tuned solutions obtained from different pre-trained checkpoints, one can easily break and even reverse this correlation. Compare, e.g., blue ($PLR = 7 \cdot 10^{-4}$) and light green ($PLR = 2 \cdot 10^{-3}$) points of fine-tuned solutions obtained with two different PLRs: with similar test error, their sharpness values differ on average by almost a factor of two.

In sum, we see no robust correlation between sharpness and generalization across fine-tuning runs from different pre-trained points, which aligns with recent work in this research area [4, 31].

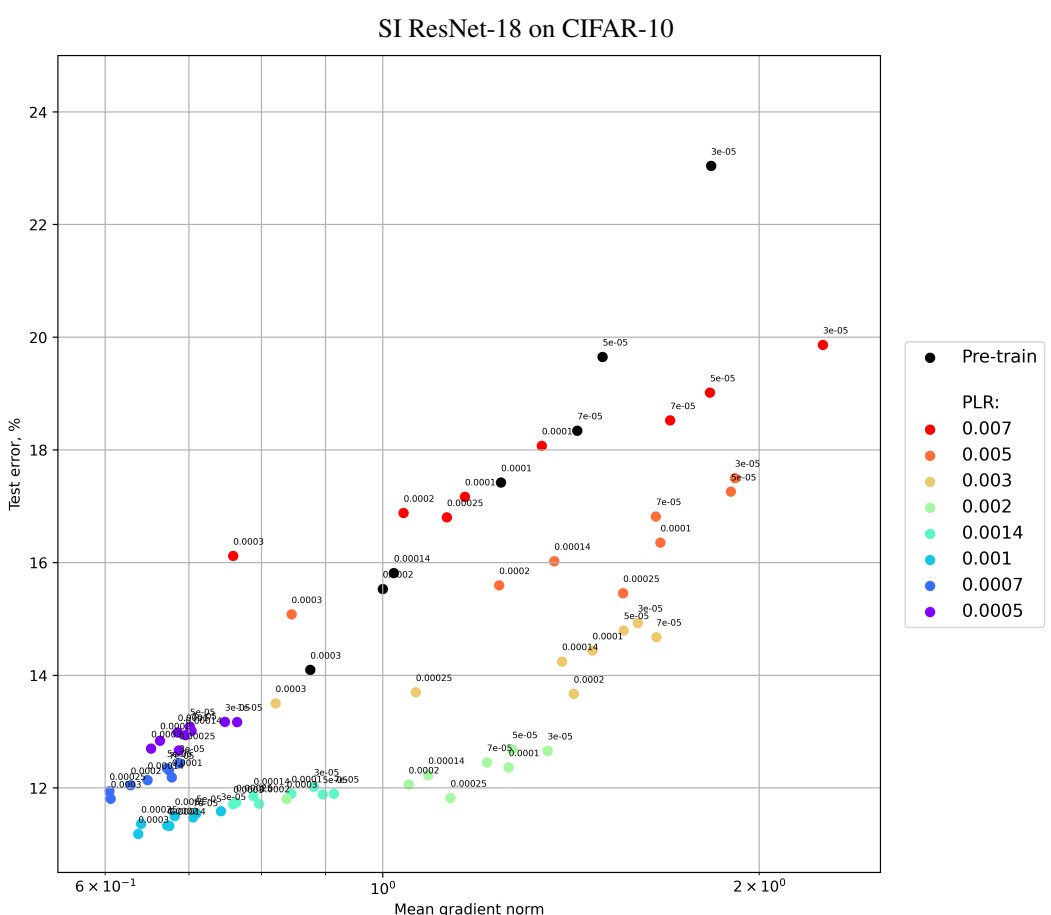

Figure 26: Scatter plot of sharpness vs. test error for the fine-tuned solutions at the same level of the training loss. Groups of points of the same color represent fine-tuned solutions with different FLRs but with the same pre-trained point. Different colors denote different PLRs of the second regime: from low (purple) to high (red). Black dots correspond to the pre-trained points of the first regime, replicating the results of Kodryan et al. [35]. SI ResNet-18 on CIFAR-10.

