# OpenReview forum: "Where Do Large Learning Rates Lead Us?"
_NeurIPS.cc/2024/Conference — NeurIPS 2024 poster_

### Official Review · Reviewer_NvNo · 2024-07-10

**Soundness:** 2
**Presentation:** 2
**Contribution:** 2
**Rating:** 5
**Confidence:** 4

**Summary:**

This paper is an empirical study focusing on the effect of large learning rates (LRs) in neural network training. The authors aim to answer two main questions:

	1	How large an initial LR is required for optimal quality?
	2	What are the key differences between models trained with different LRs?

The study reveals that only a narrow range of initial LRs slightly above the convergence threshold lead to optimal results after fine-tuning with a small LR or weight averaging. The authors observe that using LRs from this optimal range allows for the optimization to locate a basin that only contains high-quality minima.

**Strengths:**

1. The authors conduct a detailed empirical analysis on both of the above-mentioned problems in a controlled setting.

2. The study discovered that only a narrow range of initial LRs slightly above the convergence threshold lead to optimal results after fine-tuning with a small LR or weight averaging. The authors observed that using LRs from this optimal range allows for the optimization to locate a basin that only contains high-quality minima. This is a novel insight into the geometry of the loss landscape in neural network training.

3. The authors also find that initial LRs result in a sparse set of learned features, with a clear focus on those most relevant for the task.This finding contributes to our understanding of feature learning in neural networks.

**Weaknesses:**

1. The experiments in the study are limited to specific datasets (CIFARor synthetic) and neural network architectures (Resnet). Therefore, the findings may not consistently generalize to other settings. While the study offers valuable practical implications and potential explanations for widely accepted practices, these findings require further validation in more complex practical scenarios with more intricate network architectures.
2. The conclusion that “only a narrow range of initial LRs slightly above the convergence threshold lead to optimal results” is hard to apply in practice, as determining the convergence threshold beforehand is challenging and problem-specific.

**Questions:**

1. What is the appropriate definition of the “convergence threshold”? How can we determine the convergence threshold in practice without the need to traverse all possible LRs?
2. How can we practically apply the takeaway 1?
3. In Figure 3, a majority of the dots remain flat in the rightmost region. I am interested in further clarification from the authors on how they arrived at their conclusion that “both angular distances and error barriers grow as PLR increases, continuing the trend established in subregime 2B” (From Ln 239 to 240).

**Limitations:**

The authors have conducted enough discussion regarding the limitations of the paper.

---

> ### Author Rebuttal · Authors · 2024-08-07
>
> We thank the Reviewer for their thoughtful review of our work! We now address the raised concerns and questions.
>
> > The experiments in the study are limited to specific datasets (CIFARor synthetic) and neural network architectures (Resnet).
>
> We have conducted additional experiments showing that our findings transfer to other practical scenarios. Please see the general comment for more detail.
>
> > The conclusion that “only a narrow range of initial LRs slightly above the convergence threshold lead to optimal results” is hard to apply in practice, as determining the convergence threshold beforehand is challenging and problem-specific.
>
> We fully agree with this remark, however, in our general comment we also point out that precisely determining this range (subregime 2A) is often not necessary to achieve a good final solution as long as a more advanced LR schedule is used than a simple LR drop.
>
> **Answering Q1**
> The convergence threshold is understood as a value in the LR range that divides the LRs allowing the loss to converge (if trained with these fixed LR values) and the remaining LRs. In theory, finding this threshold could be done relatively efficiently using, e.g., binary search, however, as we answered above, it is not needed in practice.
> Please refer to our general comment for a more detailed discussion on this topic.
>
> **Answering Q2**
> Takeaway 1 essentially provides a recipe to obtain optimal solutions after fine-tuning with a constant small LR or weight averaging: start training with a moderately high LR from subregime 2A, a relatively narrow range above the convergence threshold. However, as we show in Appendix D, training even with significantly larger initial LRs from subregime 2B can get similar results if one chooses just a bit more complex LR schedule. Hence, we argue that for practical schedules with gradual LR decay it is often sufficient to take some reasonably large initial LR, which do not allow for convergence, to achieve good final quality.
> Please see the general comment for further discussion.
>
> **Answering Q3**
> A totally correct note, we thank the reviewer for the found inaccuracy that we will fix in the following text revision!
> What we should have written was: "In the third regime, both the angular distances and the error barriers approach their *upper limits* (angular distance of $\pi/2$, which is a typical angular distance between two independent points in high-dimensional space, and random-guess error), *completing* the trend established in subregime 2B".

---

> > ### Comment · Reviewer_NvNo · 2024-08-12
> >
> > Thank you for addressing my comments. I have decided to rise the score.

---

> > > ### Author Response · Authors · 2024-08-12
> > > **Thank you!**
> > >
> > > We are very grateful to the reviewer!
> > >
> > > We are happy that we managed to address the reviewer's comments and we believe that our discussion will benefit the future revision a lot.

---

### Official Review · Reviewer_vwNi · 2024-07-10

**Soundness:** 3
**Presentation:** 3
**Contribution:** 3
**Rating:** 6
**Confidence:** 4

**Summary:**

This paper investigates the benefits of an initial large learning rate (LR) in training neural networks. In particular, the paper identifies three training regimes in a pretraining-finetuning/model averaging context, where different regimes depend on different pre-training LRs and have distinct impacts on the fine-tuned model's final generalization capability. The paper also provides conceptual explanations for the influence of different LRs from loss landscape and feature learning perspectives and empirically justifies the explanations through controlled and real-world experiments.

**Strengths:**

In general, I enjoyed reading this paper. I think this is a solid paper in empirically analyzing the effect/benefits of an initial large learning rate in training NNs.

- The pretraining-finetuning setup considered by the paper captures the practical consideration of LR scheduling/annealing and goes beyond the fixed LR setting in prior work.

- The identified nuanced training subregimes 2A and 2B and their different impacts on fine-tuning are interesting.

- The explanation from the loss landscape perspective through linear mode connectivity is intuitive and well-justified in the experiments.

**Weaknesses:**

- Some of the paper's results have already been shown by prior work, e.g., large LRs can lead to sparse features [1].

- While the paper classifies the training into different regimes depending on the LR, how to choose LR to reach the best regime (2A) seems to remain an issue in practice: while the authors use the oracle **test** accuracy to identify different regimes, in practice one often only has access to the training loss curve (e.g., when training LLMs/large vision models, it is often not clear what constitutes an "oracle" test set to examine the model's generalization ability). It would be great if the authors could discuss whether different training regimes are also identifiable from the training loss alone.

- While the ablations are sufficient, I feel that the overall evaluation of the paper is still a bit limited, given that the paper is mainly empirical. For example, only small-scale datasets such as CIFAR-10/100 and rather small network architectures such as ResNet-18 are considered.

---

[1] SGD with large step sizes learns sparse features. ICML, 2023.

**Questions:**

- What is the recommendation of the paper in choosing the initial LR in practice?

- Why does subregime 2A reach better minima than regime 1?

- Is it possible to show similar effects on moderate-sized NNs and datasets, e.g., ResNet-50/small ViTs on Tiny ImageNet or some other datasets larger than CIFAR-10/100?

**Limitations:**

The authors have adequately addressed the limitations.

---

> ### Author Rebuttal · Authors · 2024-08-07
>
> We thank the Reviewer for their valuable feedback and overall positive assessment of our work! We respond to the comments and questions as follows.
>
> > Some of the paper's results have already been shown by prior work, e.g., large LRs can lead to sparse features [1].
>
> The results of [1] are closely related to ours, and we discuss them in Section 1.1. However, there are still important differences between this study and ours.
> The referenced work examines feature sparsity in terms of the fraction of distinct, non-zero activations at some hidden layer of a neural network over the dataset. In contrast, we study the importance of features in the *input data* for prediction. This approach allowed us to identify feature sparsity as the model preference for the most task-relevant features in the data when training with optimal initial LR values, which is a novel result. In short, our works consider different definitions of “features”: whether they are internal representations of data within a model or patterns in the input data itself.
> Furthermore, we came to a different conclusion because we found that feature sparsity (as we understand it) behaves non-monotonically w.r.t. LR, while [1] suggests a monotonic trend.
>
> > how to choose LR to reach the best regime (2A) seems to remain an issue in practice
>
> This indeed could be the case if one wanted to locate the optimal initial LRs for weight averaging or fine-tuning with a decreased LR at the end of training. However, more advanced LR schedules in practice allow for substantially higher initial LRs with similar final performance.
> Please see our general comment for further discussion.
>
> > It would be great if the authors could discuss whether different training regimes are also identifiable from the training loss alone.
>
> This is a very good point and we regret that we did not reflect it clearly in the text.
> The distinctive features of different regimes were first established in [2] based on various metrics including training loss and gradient norm (please refer to Fig. 1 in [2]). For instance, regimes 1 and 2 can be easily distinguished by the behavior of the training loss/error: whether it reaches low values (convergence) or hovers at some non-zero level. Therefore, no oracle access to the test accuracy is necessary to identify regimes.
> We will improve our wording to avoid further misinterpretations.
>
> > the overall evaluation of the paper is still a bit limited
>
> We have conducted additional experiments supporting our findings, please see the general comment for more detail.
>
> **Answering Q1**
> As we answered above, based on our findings, in practice one simply needs to choose some large enough LR value from regime 2, i.e., not allowing for convergence, and use a gradually decaying LR schedule to obtain near optimal results.
>
> **Answering Q2**
> That is a very intriguing and non-trivial question that requires further investigation. We attempted to provide some initial intuition based on the loss landscape and feature learning analysis. As we briefly discuss in Section 7, we conjecture that feature sparsity and mode proximity/linear connectivity observed in subregime 2A are closely related. We hypothesize that pre-training in this regime helps the model filter out unnecessary input features and emphasize the most relevant ones, which is beneficial for further fine-tuning/SWA. This process is dually represented in the training dynamics as finding a stable linearly connected basin of good solutions in the loss landscape. By contrast, training in regime 1 converges to the nearest minimum admissible by the chosen LR value, not allowing enough time for feature consolidation. Again, this interpretation is currently largely speculative, but deserves further study.
>
> **Answering Q3**
> As our additional experiments (in the general comment) suggest, all key findings of our work remain valid in other practical settings, including training on Tiny ImageNet and using the ViT model. Also, for example, our synthetic example, which substantially differs from the CIFAR image classification with convolutional networks, possesses all the properties of training in different regimes (Appendix G). Overall, we expect that our results can be transferred to any overparameterized training setting that allows convergence, so that regime 1 is reachable.
>
> [1] Andriushchenko Maksym et al. SGD with large step sizes learns sparse features. In International Conference on Machine Learning, 2023.
> [2] Kodryan Maxim et al.. Training scale-invariant neural networks on the sphere can happen in three regimes. Advances in Neural Information Processing Systems, 2022.

---

> > ### Comment · Reviewer_vwNi · 2024-08-08
> > **Reviewer's response**
> >
> > Thank you for your response. I appreciate the clarification and the additional experiments. I have also checked the general response and the reviews of other reviewers. At this point, I still maintain a positive rating of the paper.

---

> > > ### Author Response · Authors · 2024-08-08
> > > **Thank you!**
> > >
> > > We are grateful to the reviewer! We appreciate the feedback provided, which will certainly benefit our work.

---

### Official Review · Reviewer_Zzfe · 2024-07-11

**Soundness:** 2
**Presentation:** 4
**Contribution:** 3
**Rating:** 7
**Confidence:** 3

**Summary:**

This paper studies the effects of using initial (large) learning rates on the performance of the trained neural networks. Two key questions explored are:
1. how large are the optimal initial learning rates?
2. what's the difference between the model trained by different initial learning rates?

The paper identifies the optimal initial learning rates as a small range of learning rates slightly above the convergence threshold. Furthermore, it shows that an initial learning rate in this range locates high-quality minima and learns sparse but useful features, which other learning rates fail to do, resulting in worse generalization.

**Strengths:**

**Orginality and Significance:**

To me, understanding the effects of large learning rate neural network training is essential to understand the success of nowaday's deep learning techniques and conventions. This paper empirically answers how large the optimal (constant effective) learning rate should be in terms of its utility for future finetuning with small learning rates (or weight average training). The corresponding findings, to my best knowledge, are new and are enlightening for practical purposes. It is also a reasonable to use loss landscape geometry and feature learning capability to showcase why the corresponding learning rate can (and can not) generalize well after finetuning (or weight average training).

**Quality and Clarity:**

The paper is well written and the conclusions and findings are quite clearly presented in terms of sections and takeaways. The experiments shown in the paper are well conducted to justify the findings.

**Weaknesses:**

1. The main results are conducted in a fully scale-invariant setting with constraints on the weight norm, using projected SGD. This setup is theoretically sound but unlikely to appear in common practice. But still, the paper shows that the results derived in the controlled setting transfers to the practical setup to some extent.
2. Even though conceptually the optimal range of initial learning rates is identified as the small inverval slightly above the convergence threshold, it is still hard to parse numerically how large the learning rate should be set in practice where one is unable to explicitly find the convergence threshold.

**Questions:**

1. The conclusion in Line 219 to 220 is a bit strange to me. The previous part in this paragraph argues that finetuning with larger learning rate in this pretraining regime leads to better performance and higher-quality minima. So what does it mean by saying "unstable to fine-tuning with higher learning rates and suboptimal generalization"?
2. In Figure 3, it seems that both the training and testing error barriers in the right part of regime 1 (let's call it regime 1B) and regime 2A are nearly zero. How does this phenomena help to support the separation between the regime 1B and regime 2A, especially in terms of the loss landscape geometry? Or did I have a misunderstanding of these two figures?

**Limitations:**

The architecture and datasets tested are restrictive. The empirical findings of the paper would be more convincing if further experiments are conducted on other popular archs and datasets including VGG and imagenet, etc.

---

> ### Author Rebuttal · Authors · 2024-08-07
>
> We are grateful to the Reviewer for their positive and constructive feedback on our work! We address the concerns and questions below one by one.
>
> **Scale-invariant setting**
> As the Reviewer rightly noted, our main experiments are conducted in a specific scale-invariant setting to provide effective control over the learning rate value. Since our study is more fundamental than practical in nature, we decided to follow this setup in line with prior work examining the effect of learning rate on training dynamics. This allowed us to isolate the impact of scale invariance on the effective learning rate of the model and then extend our main findings to more practical scenarios. In our general comment, we present additional experimental results that further confirm that our claims apply to conventional training settings as well.
>
> **Finding the optimal LR range**
> It could indeed be computationally burdensome to find the exact convergence threshold for a given training setup. However, although the optimal LRs for fine-tuning with a constant small LR or weight averaging usually lie in a relatively narrow range just above this threshold, in practice even taking substantially larger initial LRs can give similar results if one chooses a slightly more complex LR schedule (Appendix D). Hence, we may conclude that for regular LR schedules it is not so important to precisely determine the convergence threshold but to choose some reasonably large initial LR above it.
> Please see our general comment for further discussion.
>
> **Answering Q1**
> Thanks for the comment, this is indeed a poor formulation, which we will correct in the next text revision!
> What we wanted to say is the following. LRs from regime 1 allow training to converge to some minima, however these minima a) have non-optimal generalization (compared to the fine-tuned/SWA solutions of subregime 2A) and b) are “unstable” in the sense that increasing LR (within the same regime 1) after convergence can knock the model out of the current minimum to a new minimum, which is perhaps better but still belongs to the minima of the first regime and therefore is not optimal.
>
> **Answering Q2**
> Indeed, local geometry, from the point of view of training/test error barriers, cannot be used to separate regimes 1 and 2A, as in fact we don’t have barriers in either case. In the first case, because all the points (pre-trained, fine-tuned, and SWA) simply coincide, which is suggested by the angular distance on the plots. In the second case, because the localized basin contains close high-quality solutions, which are linearly connected. Error barriers are, however, useful for separating subregimes 2A and 2B, as in the latter case fine-tuned/SWA solutions lose linear connectivity. At the same time, regimes 1 and 2 can be easily distinguished by the behavior of the training loss/error: whether it reaches low values (convergence) or hovers at some non-zero level (see, e.g., Figure 1 in [1]).
> We will clarify this more in the text.
>
> **Limitations**
> > The architecture and datasets tested are restrictive.
>
> We have conducted additional experiments supporting our findings, please see the general comment for more detail.
>
> [1] Kodryan Maxim et al. Training scale-invariant neural networks on the sphere can happen in three regimes. Advances in Neural Information Processing Systems, 2022.

---

> > ### Comment · Reviewer_Zzfe · 2024-08-09
> >
> > Thank you very much for your rely. I appreciate your answering to my questions, and I agree with your comments. Also the additional experiments do further demonstrate the applicability of the paper's findings in practical setups. I have no further questions and will remain my score as 7.

---

> > > ### Author Response · Authors · 2024-08-09
> > > **Thank you!**
> > >
> > > We are very grateful to the reviewer for their useful comments and high score given to our work!

---

### Author Rebuttal · Authors · 2024-08-07

We kindly thank all the reviewers for their constructive and valuable feedback that will help us further improve our paper!
We are very pleased that the reviewers assessed our findings as novel, practically important, and providing additional insights into the loss landscape geometry and feature learning in neural networks.
Below we would like to address two common issues raised in all reviews.

## Limited experimental evaluation
To show that our findings can be extrapolated to more general scenarios, we conducted additional experiments with practical ResNet-18 on Tiny ImageNet and ViT on CIFAR datasets. We attach a PDF with the corresponding plots to this comment. All results will be incorporated in the next version of the paper.

In general, our main conclusions remain the same. We can clearly observe regimes 1 and 2 (regime 3 is unstable in practical settings) as well as divide the second regime into subregimes 2A and 2B. The best fine-tuned solutions are achieved in subregime 2A, which locates a linearly connected basin and depicts a clear feature selection trend w.r.t. different frequency bands in input images. At the same time, training in regime 1 often fails to converge due to strong augmentations and exhibits catapults when fine-tuning with $FLR \gg PLR$, while fine-tuning from subregime 2B leads to diverse suboptimal solutions.

**ViT on CIFAR:** The general trends are the same as described above, although the advantage of pre-training in subregime 2A is slightly less obvious in this setup.
Notably, feature learning in transformers is different from convolutional networks, since they inherently capture lower frequencies in the data [1]. We can still see that the role of the most important features (low-frequencies in this case) grows towards 2A. Interestingly, however, the importance of midrange frequencies also peaks in subregime 2A for both datasets, which is consistent with our intuition that mid-frequencies are essential for natural image classification.
We also consider different partitions into low and midrange frequencies to answer the question of whether ViTs really rely more on the lower frequency part of the spectrum, or whether its midrange simply “starts earlier” compared to convolutional models. Figure 2 in the attached PDF shows that the latter appears to be the case. That is, if you select 5-24 or 3-24 bands for mid frequencies, then the midrange will dominate over the low-frequency bands in terms of the corresponding test accuracies.

**ResNet-18 on Tiny ImageNet:** All trends are very much alike practical experiments reported in the paper. A small drop in the pre-train accuracy for $PLR = 10^{-3}$ is due to the periodic behavior [2]: the final pre-training epoch occurs at the beginning of a period. Similar effects are reported in the paper when training ResNet-18 on CIFAR-100. That could be fixed by choosing a different random seed and/or pre-training epoch budget.

[1] Namuk Park and Songkuk Kim. How do vision transformers work? In International Conference on Learning Representations, 2021.
[2] Lobacheva Ekaterina et al. On the periodic behavior of neural network training with batch normalization and weight decay. Advances in Neural Information Processing Systems, 2021.

## What the “convergence threshold” is and how it can be used in practice
In general, by the convergence threshold (CT) for a given model and training setup we mean a learning rate (LR) value that separates regimes 1 and 2. In other words, training with a constant LR below CT leads to convergence to a minimum, while taking a larger constant LR prevents the optimization from converging. Convergence here is defined in a conventional sense, i.e., the optimized functional (training loss) closely approaches its global minimum by the end of training; in a simplified training setup without advanced data augmentations, it may be tracked by the ability of the model to fit the training data (i.e., reach ~100% training accuracy). We realize that this definition is still not constructive, and, as we show in Appendix C, CT is better understood as a small zone within the overall LR range, since the exact threshold may slightly shift depending, e.g., on the epoch budget.

However, the purpose of our work is not to quantitatively obtain the exact ranges of optimal LR values, but rather to qualitatively explore the difference between training with various LRs. The fundamental question was whether the small LRs are suboptimal to start training with, even if we ensure convergence? We indeed found that larger initial LRs, while not allowing for convergence by themselves, may lead to notably better final solutions, which is reinforced by the loss landscape and feature learning intuition. Thus, the choice of the LR influences not only the properties of the minimum achievable with this LR (as in regime 1), but the entire training process, even long before convergence, including feature learning and the optimization trajectory in the loss landscape.

Speaking of practical implications, despite our advice to take the initial LRs “slightly above the convergence threshold” for optimal fine-tuning with a constant small LR or weight averaging at the end of training, higher LRs from subregime 2B may lead to similar final quality if a more advanced LR schedule is used (Appendix D). As we discuss in part in Section 7, gradually decreasing LR can correct for an initial value that is too high. Accordingly, most practical LR schedules are designed in that manner: starting with a high value and gradually decreasing it as training progresses. Therefore, given a proper LR schedule, to achieve a good final solution in practice one essentially only needs to ensure that the initial LR value is reasonably large, i.e., it does not allow for convergence but also does not lead to numerical issues during training.

We will make more effort to elucidate these details in the next revision of the text.

---

### Decision · Program_Chairs · 2024-09-25

**Decision:**

Accept (poster)

**Comment:**

This manuscript provides a comprehensive analysis on the impact of learning rate. The main conclusion is that, only a narrow range of initial learning rates slightly above the convergence threshold can improve the generalization, and the authors provided several perspectives on the potential reason for this phenomenon. All reviewers agree the manuscript is informative, but some of the reviewers raise valid concern on it is not clear how to find this range, which should be an important future direction. I personally recommend the authors to read on a recent paper [1], which from a theoretical perspective shows larger learning rates at the beginning will lead to sparse feature. It may help the authors refine and improve the hypothesis from feature learning perspective.

[1] Qiao, Dan, et al. "Stable Minima Cannot Overfit in Univariate ReLU Networks: Generalization by Large Step Sizes." arXiv preprint arXiv:2406.06838 (2024).